



# HETEROFOR 1.0: a spatially explicit model for exploring the response of structurally complex forests to uncertain future conditions. I. Carbon fluxes and tree dimensional growth.

Mathieu Jonard[1], Frédéric André[1], François de Coligny[2], Louis de Wergifosse[1], Nicolas Beudez[2], Hendrik Davi[3], Gauthier Ligot[4], Quentin Ponette[1], Caroline Vincke[1]

[1]Earth and Life Institute, Université catholique de Louvain, Louvain-la-Neuve, 1348, Belgium
[2]Botany and Modelling of Plant Architecture and Vegetation (AMAP) Laboratory, Institut National de la Recherche Agronomique (INRA), Montpellier, 34398, France
[3]Ecologie des Forêts Méditerranéennes (URFM), Institut National de la Recherche Agronomique (INRA), Avignon, 84914, France
[4]Gembloux Agro-Bio Tech, Université de Liège, Gembloux, 5030, Belgium

*Correspondence to*: Mathieu Jonard (mathieu.jonard@uclouvain.be)





**Abstract.**

Given the multiple abiotic and biotic stressors resulting from global changes, management systems and practices must be adapted in order to maintain and reinforce the resilience of forests. Among others, the transformation of monocultures into uneven-aged and mixed stands is an avenue to improve forest resilience. To explore the forest response to these new

silvicultural practices under a changing environment, one need models combining a process-based approach with a detailed spatial representation, which is very rare.

We therefore decided to develop our own model (HETEROFOR) according to a spatially explicit approach describing individual tree growth based on resource sharing (light, water and nutrients). HETEROFOR was progressively elaborated through the integration of various modules (light interception, phenology, water cycling, photosynthesis and respiration, carbon

allocation, mineral nutrition and nutrient cycling) within CAPSIS, a collaborative modelling platform devoted to tree growth and stand dynamics. The advantage of using such a platform is to use common development environment, model execution system, user- interface and visualization tools and to share data structures, objects, methods and libraries.

This paper describes the carbon-related processes of HETEROFOR (photosynthesis, respiration, carbon allocation and tree dimensional growth) and evaluates the model performances for a mixed oak and beech stand in Wallonia (Belgium). This first

evaluation showed that HETEROFOR predicts well individual radial growth and is able to reproduce size-growth relationships. We also noticed that the more empirical options for describing maintenance respiration and crown extension provide the best results while the process-based approach best performs for photosynthesis. To illustrate how the model can be used to predict climate change impacts on forest ecosystems, the growth dynamics in this stand was simulated according to four IPCC climate scenarios. According to these simulations, the tree growth trends will be governed by the $CO_2$ fertilization effect with the

increase in vegetation period length and in water stress also playing a role but offsetting each other.



# 1 Introduction

Forest structure and composition result from soil and climate conditions, management and natural disturbances. All these drivers of forest ecosystem functioning are rapidly evolving due to global changes (Aber et al., 2001; Lindner et al., 2010; Campioli et al., 2012). While environmental and societal changes make no doubt, their magnitude and the way they will occur

locally remain largely uncertain (Lindner et al., 2014). Designing silvicultural systems and selecting tree species adapted to future conditions seems therefore a risky bet (Ennos et al., 2019). Messier et al. (2015) propose another approach recognizing that forests are complex adaptive systems whose future dynamics is inherently uncertain. To maintain the ability of forests to provide a large range of goods and services whatever the future conditions, their resilience and adaptability must be improved by favouring uneven-aged structure and tree species mixture (Thompson et al., 2009; Oliver et al., 2015). As the combinations

of site conditions, climate projections, stand structures and tree species compositions are nearly infinite, all the management options that could potentially enhance the resilience and adaptive capacity of forests cannot be tested in situ (Cantarello et al., 2017). Furthermore, such silvicultural trials would provide results only in the long run given the life span of trees. Scenario analysis based on model simulations are therefore useful to select the most promising management strategies and to evaluate their long-term sustainability. To explore forest response to new silvicultural practices and to unexperienced climate conditions

in a realistic way, one needs new process-based models able to deal with mixed and structurally complex stands and to incorporate uncertainties in future conditions (Pretzsch et al., 2015).

In connection with the traditional forestry viewing forests as a stable systems that can be controlled, many empirical models were developed to predict tree growth in monocultures considering that past conditions will remain unchanged in the future. Such models provide accurate and precise predictions of tree growth and timber yield for various thinning regimes and yield

classes (Pretzsch et al., 2008). They are however only valid for the conditions that served to develop them. On the other hand, scientists developed process-based eco-physiological models to better understand short and long-term forest ecosystem response to multiple and interacting environmental changes (Dufrêne et al., 2005). This can indeed not be done through direct experimentation because the multisite and multifactorial experiments required for doing so would be too complex and too expensive (Aber et al., 2001; Boisvenue and Running, 2006). Most experiments of environment manipulation focus on single

or few factors during a limited period of time, which precludes to properly take into account interactions, feedbacks and acclimation. To simplify the mathematical formalization of eco-physiological processes (e.g., radiation interception) and limit the calculation time, these process-based models were first designed for pure even-aged stands without considering the spatial heterogeneity of stand structure.

With the increasing interest for uneven-aged stands and tree species mixtures, cohort and tree-level models were also

developed. Pretzsch et al. (2015) reviewed 54 forest growth models to show how they represent species mixing. Among those models, 36 were process-based with 9 at the stand, 11 at the cohort and 16 at the tree level. While cohort models allow to describe the vertical structure of the stand, tree-level models are generally necessary to consider the spatial heterogeneity in the horizontal dimension. To represent stand structure in both dimensions, the model must not only operate at the individual





level but also consider the tree position. In the review of Pretzsch et al. (2015), 11 process-based models were individual-based and spatially explicit but only three of them accounted simultaneously for radiation transfer, water cycling and phenology (i.e., BALANCE, EMILION and MAESPA). Since it describes canopy and water balance processes using a state-of-the-art approach (based on a fine crown discretization), MAESPA is a very useful tool for analysing outcomes of eco-physiological

experiments (Duursma and Medlyn, 2012). MAESPA is however not suitable for multi-year simulations since it contains no routine for carbon allocation, respiration and tree dimensional growth. EMILION is also restricted to one-year simulation (no organ emergence) and is specific to pine species with a quite detailed structural approach (Bosc et al., 2000). In contrast, tree dimensional growth is well described in BALANCE which possesses a fine representation of tree structure (Grote and Pretzsch, 2002). In BALANCE, radiation interception by trees and water cycling are based on simpler eco-physiological concepts

compared to MAESPA and photosynthesis is calculated with a 10-day time step using the routine of Haxeltine and Prentice (1996). As the Forest v5.1 model (Schwalm and Ek, 2004), BALANCE has the advantage of merging two traditions, conventional growth and yield models together with process-based approaches, providing outputs familiar to foresters (classical tree and stand measurements obtained from forest inventory) as well as carbon fluxes and stocks. Among the three models, BALANCE is the only one that considers mineral nutrition through the impact of nitrogen (N) availability on tree

growth. The approach used for modelling nutrient cycling is however very simple. Soil is not partitioned into horizons and the soil chemistry processes (e.g. ion exchange, mineral weathering) are not described although they are essential to estimate bioavailability of the major nutrients other than N (P, K, Mg, Ca). Later, Simioni et al. (2016) developed the NOTG 3D model to study water and carbon fluxes in Mediterranean forests using an individual-based approach to account for the spatial structure of the stand. This model is more suited for short -term (a few years) rather than long-term (a rotation) simulations

since tree dimensions are updated based on fixed empirical relationships between diameter at breast height (*dbh*) and tree height or crown radius.

Given the lack of process-based models with detailed spatial representation, we developed a new model (HETEROFOR) using a spatially explicit approach to describe individual tree growth based on resource use (light, water and nutrients) in HETErogeneous FORrests. While the BALANCE model existed and responded roughly to our expectations, we decided to

build a new model for several reasons. First, we thought that another model of this particular type would not be redundant if based on other concepts. Instead of calculating the relative light availability, we chose to estimate radiation interception for all trees using a ray tracing approach. For calculating photosynthesis and tree transpiration, we selected a much shorter time step than in BALANCE in order to account for hourly variations in climate and soil water conditions. While we used a slightly more complex approach for the water balance module (Darcy approach instead of bucket model for soil water dynamics,

rainfall partitioning when passing through the canopy), our model rests on a simpler representation of tree structure. Second, we aimed at incorporating a detailed tree nutrition and nutrient cycling module since we realized the necessity to integrate nutritional constraints in forest growth modelling, especially for predicting the response to climate change (Fernandez-Martinez et al., 2014; Jonard et al., 2015). Finally, we wanted to develop the model in a collaborative modelling platform dedicated to tree growth and stand dynamics. Among the various platforms, CAPSIS was the only one allowing multi-model





integration and providing a user-friendly interface (Dufour-Kowalski et al., 2012). HETEROFOR was therefore progressively elaborated through the integration of various modules (light interception, phenology, water cycling, photosynthesis and respiration, carbon allocation, mineral nutrition and nutrient cycling) within CAPSIS. The advantage of such a platform is to use common development environment, model execution system, user-interface and visualization tools and to share data

5    structures, objects, methods and libraries.

To simulate the response of forests to management and changing environmental conditions, integrate and structure the existing knowledge into process-based models is essential but not sufficient. These models must also be documented and evaluated in order to know exactly their strengths and limits when analysing their outputs. The objectives of this paper are (i) to describe the carbon-related processes of HETEROFOR (photosynthesis, respiration, carbon allocation and tree dimensional growth),

10    (ii) evaluate the model ability in reconstructing tree growth in a mixed oak and beech stand of the Belgian Ardennes and compare various options for describing photosynthesis, respiration and crown extension and (iii) illustrate its potentialities by simulating tree growth dynamics in this stand under various IPCC climate scenarios. As the whole model could not be presented in the same paper, the other aspects will be described in companion papers.



## 2. Materials and methods

### 2.1 Overall operation of the HETEROFOR model

HETEROFOR is a model integrated in the CAPSIS platform dedicated to forest growth and dynamics modelling (Dufour-Kowalski et al., 2012). CAPSIS provides to HETEROFOR the execution system and the methods necessary to run simulations

and display the results. When running simulations with HETEROFOR, CAPSIS creates a new project in which the variables describing the forest state are stored at a yearly time step, starting from the initial forest characteristics (initial step). Though some data structures and methods are shared with other models integrated in CAPSIS, the initialisation and evolution procedures are specific to HETEROFOR.

For the initialization, HETEROFOR loads a series of files containing tree species parameters, input data on tree (location,

dimensions and chemistry), soil (chemical and physical properties) and open field hourly meteorological data. These data are used to create trees and soil horizons at the initial step. Then, HETEROFOR predicts tree growth at a yearly time step based on underlying processes modelled at finer time steps and at different spatial levels.

After the initialization step, and at the end of each successive yearly time step, the phenological periods for each deciduous species (leaf development, leaf colouring and shedding) are defined for the next step from meteorological data. When no

meteorological measurements are available, the vegetation period is defined by the user who provides the budburst and the leaf shedding dates. Knowing the key phenological dates and the rates of leaf expansion, colouring and falling, the foliage state of the deciduous species is predicted at any time during the year and is used to carry out a radiation budget with the SAMSARALIGHT library of CAPSIS (Courbaud et al., 2003).

Based on a ray tracing approach, SAMSARALIGHT calculates the solar radiation absorbed by the trunk and the crown of each

individual tree and the radiation transmitted to the ground. This allows HETEROFOR to estimate the proportions of incident radiation absorbed by the trunk and the crown of each tree and the part transmitted to the ground either on average over the whole vegetation period (simplified budget) or hourly for several key dates (detailed budget). These proportions and the incident radiation measured in the meteorological station are used during the next step to compute the hourly global, direct and diffuse radiation absorbed per unit bark or leaf area. Predicting how solar energy is distributed within the forest ecosystem

is necessary to estimate foliage, bark and soil evaporation, tree transpiration and leaf photosynthesis.

Every hour, HETEROFOR performs a water balance and updates the water content of each horizon. Rainfall is partitioned in throughfall, stemflow and interception (Andre et al., 2008a; 2008b and 2011). Part of the rainfall reaches directly the ground (throughfall) while the rest is intercepted by foliage and bark. These two tree compartments both have a certain water storage capacity which is regenerated by evaporation. When the foliage is saturated, the overflow joins the throughfall flux whose

proportion increases. As the bark saturates, water flows along the trunk to form stemflow. Foliage and bark storage capacity as well as stemflow proportion are determined at the tree level and then upscaled to the stand level, while evaporation from these surfaces is evaluated at the stand scale. Throughfall is also determined at the stand level as the difference between incident rainfall and the abovementioned fluxes. Throughfall and stemflow supply the first soil horizon (forest floor) with water while



soil evaporation and root uptake deplete it. The water evaporation from the soil (as well as from the foliage and the bark) is calculated at stand scale with the Penman-Monteith equation. Using the same equation, individual tree transpiration is estimated by determining the stomatal conductance from tree characteristics, soil extractable water and meteorological conditions. The distribution of root water uptake among the soil horizons is done according to the water accessibility (evaluated

based on the water potential and the vertical distribution of fine roots). Water exchanges between soil horizons are considered as water inputs (capillary rise) or outputs (drainage). This soil water transfers are calculated based on the water potential gradients according to the Darcy law and using pedotransfer functions to determined soil hydraulic properties. All these soil water fluxes are considered at the stand level.

The gross primary production of each tree (*gpp*) is either obtained based on a radiation use efficiency approach distinguishing

sunlit and shaded leaves or calculated hourly using the Farquhar et al. (1980) model. The latter is analytically coupled to the stomatal conductance model proposed by Ball et al. (1987). The photosynthesis is computed using the Library CASTANEA also present in CAPSIS (Dufrêne et al., 2005). This calculation requires the proportions of sunlit and shaded leaves, the direct and diffuse photosynthetically active radiation (*PAR*) absorbed per unit leaf area and the relative extractable water reserve (REW). *gpp* is then converted to net primary production (*npp*) after subtraction of growth and maintenance respiration.

Maintenance respiration is either considered as a proportion of *gpp* (depending on the crown to stem diameter ratio) or calculated for each tree compartment by considering the living biomass, the nitrogen concentration and a Q10 function for the temperature dependency following Ryan (1991) as in Dufrêne et al. (2005). Carbon allocation is made in priority to foliage and fine roots by ensuring a functional balance between carbon fixation and nutrient uptake through a fine root to leaf biomass ratio depending on the tree nutritional status (Helmisaari et al., 2007). Allometric relationships are then used to describe carbon

allocation to structural components (trunk, branches and structural roots) and to derive tree dimensional growth (diameter at breast height, total height, height to crown base, height of largest crown extension, crown radii in 4 directions) while considering competition with neighbouring trees (Fig. 1).

Knowing the chemical composition of the tree compartments for a given tree nutrient status, HETEROFOR computes the individual tree nutrient requirements based on the estimated growth rate and deduces the tree nutrient demand after subtraction

of the amount of re-translocated nutrient. On another hand, the potential nutrient uptake is obtained by calculating the maximum rate of ion transport towards the roots (by diffusion and mass flow). The actual uptake is then determined by adjusting the tree nutrient status and growth rate so that tree nutrient demand matches soil nutrient supply. The nutrient limitation of tree growth is achieved through the regulation of maintenance respiration and through the effect of the tree nutrient status on fine root allocation.

The central compartment of the nutrient cycling is the soil solution whose chemical composition is in equilibrium with the exchange complex and the secondary minerals. This compartment receives the nutrients coming from atmospheric deposition, organic matter mineralization and primary mineral weathering, and is depleted by root uptake and immobilization in micro-organisms. The chemical equilibrium within the soil solution, with the exchange complex or the minerals is updated yearly



with the PHREEQC geochemical model (Charlton and Parkhurst, 2011) coupled to HETEROFOR through a dynamic link library.

In this paper, we present a detailed description of the processes regulating the carbon fluxes (Fig. 1) while the coupling with the radiation transfer library (SAMSARALIGHT), the phenology module, the water balance module and the nutrient cycling and tree nutrition module will be described in details in other papers.

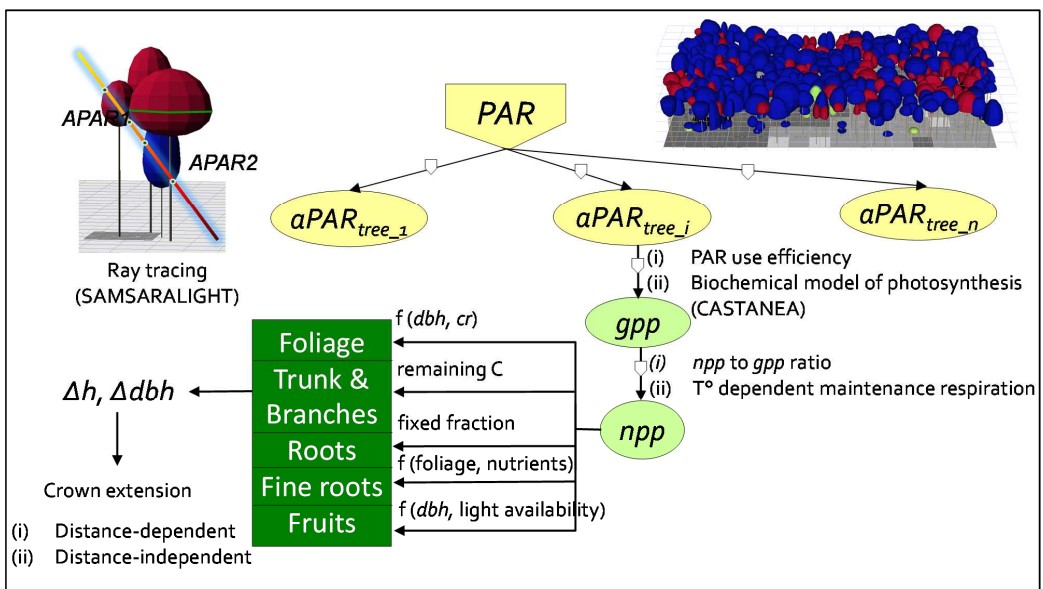

**Figure 1. Conceptual diagram of the HETEROFOR model. The incident *PAR* radiation is absorbed by individual trees using a ray tracing model (SAMSARALIGHT library). Then, the absorbed *PAR* (*aPAR*) is converted into gross primary production (*gpp*) based on the *PAR* use efficiency concept or with a biochemical model of photosynthesis (coupling with the CASTANEA library). The net primary production (*npp*) is obtained using a *npp* to *gpp* ratio or by subtracting the growth and maintenance respiration (the latter being temperature dependent). *npp* is first allocated to foliage using an allometric equation function of tree diameter (*dbh*) and crown radius (*cr*). The carbon allocated to fine roots is determined based on a fine root to foliage ratio dependent on the tree nutritional status. Fruit production is calculated with an allometric equation based on *dbh* and on light availability. The remaining carbon is allocated to structural organs (roots, trunk and branches) using a fixed proportion for the below-ground part. *dbh* and height growth (*Δdbh, Δh*) are deduced from the change in aboveground biomass by deriving and rearranging an allometric equation. Finally, crown extension is predicted with a distance-dependent or -independent approach.**





## 2.2 Detailed model description

### 2.2.1 Initialization

To initialize HETEROFOR, the relative position ($x$, $y$, $z$) and the main dimensions of each tree must be provided: girth at breast height (*gbh* in cm), height (*h* in m), height of maximum crown extension (*hlce* in m), height to crown base (*hcb* in m) and
crown radii in the four cardinal directions (*cr* in m). During the initialization phase, the biomass of each tree compartment is calculated according to the equations used for carbon allocation (see sect. 2.2.4). If available, site-specific allometric equations can also be used to calculate initial biomasses of tree compartments. When data on fruit litterfall are available, a file providing the amount of fruit litterfall per year and per tree species can be loaded and used to adapt the allometric equations predicting fruit production at the individual level. When the water balance module is activated, two additional files must be loaded: a file
describing soil horizon properties and another one for the hourly meteorology.

### 2.2.2 Gross primary production

The annual gross primary production of each tree (*gpp* in kgC yr$^{-1}$) is calculated either based on a *PAR* use efficiency (*PUE*) approach (Monteith, 1977) or using the photosynthesis method of the CASTANEA model (Dufrêne et al., 2005). Whatever the option retained, a series of variables are needed to calculate *gpp*.

For the *PUE* approach, the model uses the solar radiation absorbed by each tree during the vegetation period (*aRAD* in MJ yr$^{-1}$). *aRAD* is then converted in *PAR* (*aPAR* in mol photons yr$^{-1}$) by supposing that 46% of the solar radiation (*RAD*) is *PAR* and that 1 MJ is equivalent to 4.55 moles of photons. The diffuse and direct components of *aPAR* are also considered (*aPAR$_{diff}$* and *aPAR$_{dir}$* in mol photons yr$^{-1}$). While all the leaves receive diffuse *PAR*, only sunlit leaves absorb direct *PAR*. To estimate the sunlit leaf proportion (*Prop$_{sl}$*) at the tree level, HETEROFOR uses an adaptation of the classical stand-scale approach based
on the Beer-Lambert law (The, 2006):

$$Prop_{sl} = \frac{1-\exp(-k \cdot LAI)}{k} \tag{1}$$

with

$k$, the extinction coefficient (m$^{-1}$),

$LAI$, the leaf area index (m² m$^{-2}$).

At the individual scale, the leaf area index is calculated by dividing the tree leaf area (*a$_{leaf}$* in m²) by the crown projection area (*cpa* in m²). The value obtained is then multiplied by the light competition index (*LCI* in MJ MJ$^{-1}$) to account for the shading effect of the neighbouring trees:

$$Prop_{sl} = \frac{1-\exp\left(-k \cdot \frac{a_{leaf}}{cpa}\right)}{k} \cdot LCI \tag{2}$$

where LCI is the ratio between the absorbed radiation calculated with and without neighbouring trees in
SAMSARALIGHT. LCI ranges from 1 (no light competition) to 0 (no light reaching the tree).



To adapt the *PAR* use efficiency concept (*PUE*) at the tree level, we considered a distinct *PUE* for sunlit (*sl*) and shaded (*sh*) leaves and calculated an average *PUE* weighted as follows:

$$pue = \frac{aPAR_{diff} \cdot (Prop_{sl} \cdot PUE_{sl} + Prop_{sh} \cdot PUE_{sh}) + aPAR_{dir} \cdot PUE_{sl}}{aPAR} \tag{3}$$

This *pue* is then used to calculate *gpp* based on *aPAR* and a reducer accounting for water stress ($red_{water}$):

$$gpp = aPAR \cdot pue \cdot red_{water} \tag{4}$$

The default value of $red_{water}$ is 1 but, when the hydrological module is activated, it is set to the ratio between the actual and the potential (i.e., considering no soil water limitation) tree transpiration ($t_{actual}$ and $t_{pot}$, in l per year). This ratio estimates the fraction of the vegetation period during which stomata are partially or totally closed due to limitation in soil water availability. Since this ratio is always lower or equal to 1, a correction factor is applied to avoid introducing a bias.

$$red_{water} = \frac{t_{actual}}{t_{pot}} \cdot corr \tag{5}$$

*gpp* can also be estimated using the photosynthesis method of CASTANEA (Dufrêne et al., 2005). This method consists in the biochemical model of Farquhar et al. (1980) analytically coupled with the approach of Ball et al. (1987) that linearly relates stomatal conductance to the product of the carbon assimilation rate by the relative humidity. The slope of this relationship varies with the soil water availability characterized in HETEROFOR based on the relative extractable water (see de Wergifosse et al., in prep). The formulation of Ball et al. (1987) was slightly adapted to the tree level by accounting for the influence of tree height. Indeed, leaf water potential increases with leaf height and induces a decrease in stomatal conductance (Ryan and Yoder, 1997; Schäfer et al., 2000).

The photosynthesis method requires, at an hourly time step, the direct and diffuse *PAR* absorbed per unit leaf area. The direct *PAR* is intercepted only by sunlit leaves and is obtained by multiplying the hourly incident *PAR* (μmol photons m$^{-2}$ s$^{-1}$) by the proportion of direct *PAR* absorbed by sunlit leaves. For a tree, this proportion is by default fixed for the whole vegetation period and calculated as the ratio between the direct *PAR* absorbed per unit sunlit leaf area during the vegetation period (in mol photons.m$^{-2}$.yr$^{-1}$) and the incident *PAR* cumulated over the same period (in mol photons m$^{-2}$ yr$^{-1}$). A similar procedure is used for the diffuse absorbed *PAR*, except that it is related to the total leaf area. When using the detailed version of SAMSARALIGHT, the proportions of direct/diffuse *PAR* absorbed per unit leaf area change every hour during the day and depending on the phenological stage.

### 2.2.3 Growth and maintenance respiration

*gpp* is converted to annual net primary production (*npp* in kgC yr$^{-1}$) using either a ratio depending on the crown to stem diameter ratio (Eq. 6) or after subtraction of growth (*gr*) and maintenance respiration (*mr*) (Eq. 7) according to the theory of respiration developed by Penning de Vries (1975).

$$npp = gpp \cdot r_{npp\_gpp}(DdI) \tag{6}$$

$$npp = gpp - mr - gr \tag{7}$$





Mäkelä and Valentine (2001) showed that the *npp* to *gpp* ratio changes with tree size. Based on simulated *gpp* and *npp* reconstructed by using the model in reverse mode (see sect. 2.2.7), we tested the impact of several variables characterizing tree size (height, *dbh*, crown radius, crown volume, crown to stem diameter ratio, aboveground volume or biomass) on the *npp* to *gpp* ratio. The best relationship was obtained with the crown to stem diameter ratio (*Dd* in m m$^{-1}$) which had a negative

5  effect on the *npp* to *gpp* ratio. This indicates that the proportion of *gpp* lost by respiration increases for trees with a large crown. As the crown to stem diameter ratio changes during the course of the tree development for some tree species, we standardized it to obtain a crown to stem diameter index (*DdIndex*).

$$r_{npp\_gpp} = \alpha + \beta \cdot DdIndex \tag{8}$$

where $\alpha$ and $\beta$ are parameters and *DdIndex* is defined as :

$$DdIndex = \frac{Dd}{Dd_{pred}} \tag{9}$$

with

*Dd*, the crown to stem diameter ratio determined from the tree mean crown radius ($cr_{mean}$ in m) and diameter at breast height (*dbh* in m),

$Dd_{pred}$, the crown to stem diameter ratio predicted based on the girth at breast height (*gbh* in cm):

$$Dd_{pred} = \alpha + \beta \cdot gbh + \gamma \cdot \frac{1}{gbh} + \delta \cdot \frac{1}{gbh^2} \tag{10}$$

In Eq. (7), maintenance respiration is calculated for each tree by summing the maintenance respiration of each organ estimated from the nitrogen content of its living biomass and considering a $Q_{10}$ function for the temperature dependency. During daytime, the inhibition of foliage respiration by light is taken into account by considering that this inhibition reduces respiration by 62% (Villard et al., 1995).

$$mr = \sum_{organ} \left( b_{organ} \cdot f_{living} \cdot [N] \cdot R_{T_{ref}} \cdot Q_{10\_organ}^{\frac{T-T_{ref}}{10}} \right) \tag{11}$$

with

$b_{organ}$, the organ biomass (kg of organic matter),

$f_{living}$, the fraction of living biomass,

[N], the nitrogen concentration (g kg$^{-1}$),

25  $R_{T_{ref}}$, the maintenance respiration per g of N at the reference temperature (15°C),

T, is the air temperature for aboveground organs or the soil horizon temperature for roots (see Appendix A). Root maintenance respiration is estimated for each soil horizon separately.

The fraction of living biomass is fixed to 1 for leaves and fine roots or equals the proportion of sapwood for the structural organs. The sapwood proportion is derived from the sapwood area ($a_{sapwood}$ in cm²) determined based on an empirical

30  function of organ diameter ($\emptyset_{organ}$ in cm):

$$a_{sapwood} = a + b \cdot \emptyset_{organ} + c \cdot \emptyset_{organ}^2 \tag{12}$$





Growth respiration is the sum of the organ growth respiration which is proportional to the organ biomass increment (see sect. 2.2.4):

$$gr = \sum_{organ}(R_{gr} \cdot \Delta b_{organ}) \qquad (13)$$

where $R_{gr}$ is the growth respiration per unit biomass increment (kgC kgC$^{-1}$).

### 2.2.4 Carbon allocation and dimensional growth

For each tree, the *npp* and the carbon retranslocated from leaves and roots ($rt_{leaf}$ and $rt_{fine\ root}$ in kgC yr$^{-1}$) are distributed among the various tree compartments at the end of the year. $rt_{leaf}$ and $rt_{fine\ root}$ are determined as follows :

$$rt_{leaf\ or\ fine\ root} = b_{leaf\ or\ fine\ root} \cdot \delta_{leaf\ or\ fine\ root} \cdot rtr_{leaf\ or\ fine\ root} \qquad (14)$$

where $b_{leaf}$ and $b_{fine\ root}$ are the tree leaf and fine root biomasses (kgC), $\delta_{leaf}$ and $\delta_{fine\ root}$ are the leaf and fine root turnover rates (kgC kgC$^{-1}$ yr$^{-1}$), and $rtr_{leaf}$ and $rtr_{fine\ root}$ are the leaf and fine root retranslocation rates (kgC kgC$^{-1}$).

$b_{leaf}$ is estimated with an allometric equation based on the stem diameter at breast height (*dbh* in cm) and on the crown to stem diameter ratio (*Dd*):

$$b_{leaf} = \alpha \cdot dbh^{\beta} \cdot Dd^{\gamma} \qquad (15)$$

$b_{fine\ root}$ is deduced from the leaf biomass using the fine root to leaf ratio ($r_{fine\ root\ to\ foliage}$):

$$b_{fine\ root} = b_{leaf} \cdot r_{fine\ root\_leaf} \qquad (16)$$

$r_{fine\ root\_leaf}$ takes a value between a minimum ($r_{fine\ root\_leaf\_min}$) and maximum ($r_{fine\ root\_leaf\_max}$) ratio depending on the tree nutritional status, in accordance with the concept of functional balance (Mäkela 1986). This means that a higher ratio is used (more carbon allocation to fine roots) when tree suffers from nutrient deficiency. For each nutrient, a candidate ratio is obtained based on a linear relationship depending on the nutritional status. The ratio increases when the nutritional status deteriorates and this effect is more pronounced for nitrogen (N) > phosphorus (P) > potassium (K) > magnesium (Mg) > calcium (Ca). Among the candidate ratios, the maximum is retained. For each nutrient, the nutritional status is bounded between 0 and 1 and calculated based on the foliar concentrations (provided in the inventory file) and on the optimum and deficiency thresholds (Mellert and Göttlein, 2012).

$$Status(Nutrient) = \frac{[Foliar\ Nutrient] - Deficiency}{Optimum - Deficiency} \qquad (17)$$

The leaf and fine root litter amounts ($s_{leaf}$ and $s_{fine\ root}$ in kgC yr$^{-1}$) are estimated based on the turnover rate taking into account the retranslocation:

$$s_{leaf\ or\ fine\ root} = b_{leaf\ or\ fine\ root} \cdot \delta_{leaf\ or\ fine\ root} \cdot (1 - rt_{leaf\ or\ fine\ root}) \qquad (18)$$

In the allocation, priority is given to leaves and fine roots. The carbon allocated to leaves corresponds to the annual leaf production ($p_{leaf}$ in kgC yr$^{-1}$) which is equal to the amount of leaves fallen the previous year plus the leaf biomass change ($\Delta b_{leaf}$ in kgC yr$^{-1}$):





$$p_{leaf} = b_{leaf_{t-1}} \cdot \delta_{leaf} + \Delta b_{leaf} \tag{19}$$

where $\Delta b_{leaf}$ is determined by :

$$\Delta b_{leaf} = b_{leaf_t} - b_{leaf_{t-1}} \tag{20}$$

with $b_{leaf_{t-1}}$ and $b_{leaf_t}$ being the tree leaf biomasses corresponding to the previous and the current years, respectively.

The fine root production is then estimated according to the same logic:

$$p_{fine\ root} = b_{fine\ root_{t-1}} \cdot \delta_{fr} + \Delta b_{fr} \tag{21}$$

where $b_{fine\ root_{t-1}}$ is provided by Eq. (16).

When the carbon allocated to leaf and fine root is higher than the *npp* plus the retranslocated carbon (suppressed trees with low *gpp* and *npp* for their size), the leaf and fine root productions are recalculated so that they do not exceed 90% of the available carbon.

Then, the fruit production ($p_{fruit}$ in kgC yr$^{-1}$) is estimated with an allometric equation similar to Eq. (15) and is considered directly proportional to the light competition index. A threshold *dbh* ($dbh_{threshold}$ in cm) is fixed below which no fruit production occurs.

$$p_{fruit} = \alpha \cdot LCI \cdot (dbh - dbh_{threshold})^{\beta} \tag{22}$$

Part of the carbon is also used to compensate for branch and root mortality. The branch mortality ($s_{branch}$ in kgC yr$^{-1}$) is described with an equation of the same form as Eq. (15) while the structural root mortality ($s_{root}$ in kgC yr$^{-1}$) is obtained using a turnover rate similar to that of the branches.

After subtracting the leaf, fine root and fruit productions and the root and branch senescences, the remaining carbon is allocated to structural organ growth:

$$\Delta b_{structural} = npp + rt - p_{leaf} - p_{fine\ root} - p_{fruit} - s_{branch} - s_{root} \tag{23}$$

At this stage, the remaining carbon is partitioned between the above- and below-ground parts of the tree according to a fixed root to shoot ratio ($r_{root\_shoot}$):

$$\Delta b_{structural\_above} = \frac{\Delta b_{structural}}{(1 + r_{root\_shoot})} \tag{24}$$

$$\Delta b_{structural\_below} = \Delta b_{structural} - \Delta b_{structural\_above} \tag{25}$$

The increment in aboveground structural biomass is then used to determine the combined increment in *dbh* and total height (*h* in m) based on an allometric equation used to predict aboveground woody biomass (Genet et al., 2011; Hounzandji et al., 2015):

$$b_{structural\_above} = \alpha + \beta (dbh^2 \cdot h)^{\gamma} \tag{26}$$

Deriving this equation and rearranging terms gives:

$$\Delta b_{structural\_above} = \beta \gamma (dbh^2 \cdot h)^{\gamma - 1} \Delta (dbh^2 \cdot h) \tag{27}$$





$$\Delta( dbh^2 \cdot h) = \frac{\Delta b_{structural\_above}}{\beta \gamma ( dbh^2 \cdot h)^{\gamma-1}} \tag{28}$$

The development of the left term provides:

$$\Delta( dbh^2 \cdot h) = (dbh + \Delta dbh)^2 \cdot (h + \Delta h) - dbh^2 \cdot h \tag{29}$$

which can be further developed (see Appendix B for details) to isolate $\Delta h$:

$$\Delta h \cong \frac{\Delta( dbh^2 \cdot h)}{dbh^2} - \frac{h \cdot \Delta dbh^2}{dbh^2} \tag{30}$$

From Eq. (30), we know that the height increment can be expressed as a function of $\frac{\Delta( dbh^2 \cdot h)}{dbh^2}$. In the following, we refer to it as the height growth potential ($\Delta h_{pot}$) since it corresponds to the height increment if all the remaining carbon was allocated to height growth. Contrary to the other term of Eq. (30) $\left(\frac{h \cdot \Delta dbh^2}{dbh^2}\right)$ which is unknown, this height growth potential can be evaluated at this step by dividing the result of Eq. (28) by $dbh^2$. However, depending on the level of competition for light and on the tree size, only part of this height growth potential will be effectively realised for height increment. For each tree species, an empirical relationship predicting height growth from the height growth potential, the light competition index and the tree size ($dbh$ or height) was therefore fitted based on successive inventory data (see Appendix E):

$$\Delta h = a + b \cdot dbh + c \cdot h + d \cdot LCI + e \cdot \Delta h_{pot} + f \cdot \left(\Delta h_{pot}\right)^2 + g \cdot \left(\Delta h_{pot}\right)^3 \tag{31}$$

The $dbh$ increment is then determined by rearranging Eq. (29):

$$\Delta dbh = \sqrt{\frac{\Delta( dbh^2 \cdot h) + dbh^2 \cdot h}{(h + \Delta h)}} - dbh \tag{32}$$

The increments in root, stem and branch biomass are obtained as follows:

$$\Delta b_{root} = r_{root\_shoot} \cdot \Delta b_{structural\_above} \tag{33}$$

$$\Delta b_{stem} = f \cdot \rho \cdot ((dbh + \Delta dbh)^2 \cdot (h_{del} + \Delta h_{del}) - dbh^2 \cdot h_{del}) \tag{34}$$

$$\Delta b_{branch} = \Delta b_{structural\_above} - \Delta b_{stem} \tag{35}$$

with

$f$ is the form coefficient (m³ m⁻³),

$\rho$ is the stem volumetric mass (kgC m⁻³),

$h_{del}$ is the Delevoy height (m) corresponding to the height at which stem diameter is half the diameter at breast height (see Appendix C).

The branch and root biomasses are then distributed in 3 categories defined based on the diameter: small branches/roots < 4 cm, medium branches/roots between 4 and 7 cm, coarse branches/roots > 7 cm. The proportions of small, medium and coarse branches/roots are determined based on the equations of Hounzandji et al. (2015).





### 2.2.5 Crown extension

Depending on whether the competition with the neighbouring trees is taken into account or not, the crown dynamics can be describe by two different approaches. When local competition is not considered (distance-independent approach), change in crown dimensions are derived from *dbh* or height increment based on empirical relationships:

$$\Delta hlce = hlce\% \cdot \Delta h \tag{36}$$

$$\Delta hcb = hcb\% \cdot \Delta h \tag{37}$$

$$\Delta cr = Dd_{pred} \cdot \frac{\Delta dbh}{200} \tag{38}$$

where $hcb\%$ and $hlce\%$ are the proportions of the total height corresponding to the height to crown base ($hcb$ in m) and to the height of largest crown extension ($hlce$ in m), respectively;

$\Delta cr$ is the change in crown radius (in m) whatever the direction;

$Dd_{pred}$ is the crown to stem diameter ratio estimated by Eq. (10).

Alternatively, the changes in crown dimensions can be described based on the competition with the neighbouring trees (distance-dependent approach). The space around a target tree is divided into 4 sectors according to the 4 cardinal directions (North between 315° and 45°, East between 45° and 135°, South between 135° and 225°, West between 225° and 315°). In each sector, the tree which is the closest to the target tree is retained as a competitor if its height is higher than the $hcb$ of the target tree. Beyond a certain distance (i.e., two times the maximal crown radius: 10 m), no competitor is considered. For each main direction, the model calculates an $hlce$ at equilibrium ($hlce_{eq}$ in m) for the target tree. This $hlce$ at equilibrium is located between a minimum ($hcb$ in m) and a maximum ($hlce_{max}$ in m). $hlce_{max}$ is obtained by determining the higher intersection between the potential crowns of the target tree and the competitor. The potential crown of a tree is the crown that this tree would have had in absence of competition and is considered as having the shape of a half ellipsoid centred on the tree trunk and with the semi-axis lengths equal to the tree potential crown radius ($cr_{pot}$ in m, see below) and to the crown length ($h - hcb$). $hlce_{eq}$ is positioned between the minimum and the maximum values according to the competition intensity estimated based on the target tree and the competitor heights ($h_{target}$ and $h_{comp}$ in m) as well as the $hcb$ of the target tree (Appendix D):

$$hlce_{eq} = hcb + (hlce_{max} - hcb) \cdot max\left(0, min\left(1, \frac{h_{comp} - hcb}{h_{target} - hcb}\right)\right) \tag{39}$$

The four values of $hlce_{eq}$ are then averaged ($hlce_{eq\_mean}$).

Finally, the change in $hlce$ is determined as follows:

if $hlce < hlce_{eq\_mean}$,

$$\Delta hlce = min(\Delta hlce_{max}, hlce_{eq\_mean} - hlce) \tag{40}$$

else,

$$\Delta hlce = max(-\Delta hlce_{max}, hlce_{eq\_mean} - hlce) \tag{41}$$

where $\Delta hlce_{max}$ is the maximum change in $hlce$ allowed by the model.





The change in $hcb$ is obtained with the same logic:

if $hcb < hcb_{eq\_mean}$,

$$\Delta hcb = \min(\Delta hcb_{max}, hcb_{eq\_mean} - hcb) \tag{42}$$

else,

$$\Delta hcb = \max(-\Delta hcb_{max}, hcb_{eq\_mean} - hcb) \tag{43}$$

where $hcb_{eq\_mean}$ is the $hcb$ estimated from the tree height based on $hcb\%$ (Eq. 37).

The change in the four crown radii is calculated based on crown radii at equilibrium ($cr_{eq}$ in m) which are estimated by considering the competitive strength of the target and neighbouring trees. For a given direction, $cr_{eq}$ is calculated based on the potential (free growth) crown radius of the target tree ($cr_{pot\_target}$ in m) and of its competitor ($cr_{pot\_comp}$ in m), the distance between the two trees ($d$ in m) and the crown overlap ratio ($r_{overlap}$ in m m$^{-1}$):

$$cr_{eq} = \frac{cr_{pot\_target}}{cr_{pot\_target} + cr_{pot\_comp}} \cdot d \cdot r_{overlap\_target} \tag{44}$$

The potential crown radius ($cr_{pot}$) of a tree if determined by:

$$cr_{pot} = \frac{dbh}{200} \cdot Dd_{pred} \cdot sh \tag{45}$$

where $Dd_{pred}$ is the crown to stem diameter ratio estimated by Eq. (10) and $sh$ is a coefficient allowing to shift from the mean to the maximum $Dd_{pred}$.

The crown overlap ratio is estimated by considering neighbouring trees of the same species two by two and by calculating the ratio between the sum of their crown radii and the distance between the corresponding tree stems. This overlap ratio accounts for the capacity of a tree species to penetrate in neighbouring crowns.

The change in crown radius is then determined as follows for each direction:

if $cr < cr_{eq}$,

$$\Delta cr = \min(\Delta cr_{max}, cr_{eq} - cr) \tag{46}$$

else,

$$\Delta cr = \max(-\Delta cr_{max}, cr_{eq} - cr) \tag{47}$$

with $\Delta cr_{min}$ and $\Delta cr_{max}$ being respectively the minimum and the maximum change in $cr$ allowed by the model. They are obtained similarly as $cr_{pot}$:

$$\Delta cr_{max} = \frac{\Delta dbh}{200} \cdot Dd \cdot sh \tag{48}$$





### 2.2.6 Tree harvesting and mortality

During the simulation, thinning can be achieved at each annual step either (i) by selecting the trees from a list or a map or according to tree characteristics (tree species, age, *dbh*, height,…), or (ii) by defining the number of trees to be thinned per diameter class using an interactive histogram, or (iii) by loading a file listing the trees that must be thinned. In addition, the

thinning methods developed for GYMNOS and QUERGUS are compatible with HETEROFOR. They allow to reach a target basal area, density or relative density index by thinning from below or from above or by creating gaps (Ligot et al., 2014).

When the *npp* of a tree is not sufficient to ensure a normal leaf and fine root development (for suppressed trees and/or after a severe drought), the leaf biomass is reduced and induces a defoliation which is estimated as follows:

$$Def = \frac{b_{leaf} - b_{leaf\_corr}}{b_{leaf}} \cdot 100 \qquad (49)$$

10       where $b_{leaf}$ and $b_{leaf\_corr}$ are respectively the leaf biomass estimated with Eq. (15) and the leaf biomass corrected to match the available carbon (see sect. 2.2.4).

Tree mortality occurs when trees reach a defoliation of 90%, considering that a tree with less than 10% of its leaves will never recover. Hence, HETEROFOR takes into account the mortality resulting from carbon starvation due to light competition and/or water stress (stomatal closure).

### 2.2.7 Growth reconstruction

HETEROFOR was adapted to allow the user to run it in reverse mode starting from the known increments in *dbh* and *h* to reconstruct individual *npp* using exactly the same parameters and equations as in the normal mode. To achieve a reconstruction, an inventory file with tree measurements must be loaded to create the initial step. From this initial step, the reconstruction tools

can be launched and requires another inventory file with tree measurements achieved one or several years later. Based on these two inventories, HETEROFOR calculates the mean *dbh* and *h* increments for each tree and use the model equations to reconstruct each step and evaluate among other individual *npp*.

### 2.3 Input variables and parameter setting for a case study

The model was tested using data from the Baileux site located in the western part of the Belgian Ardennes at 300 m elevation (50° 01' N, 4° 24' E). The average annual rainfall is slightly above 1000 mm and the mean annual temperature is 8°C. The forest (60 ha) consists of sessile oak (*Quercus petraea* LIEBL.) and European beech (*Fagus sylvatica* L.) and lies on acid brown earth soil (luvisol according to the FAO soil taxonomy) with a moder humus and an $A_hB_wC$ profile. The soil has been developed on a loamy and stony solifluxion sheet in which weathering products of the bedrock (Lower Devonian: sandstone

and schist) were mixed with added periglacial loess.





By the end of the 19th century, the forest was probably an oak coppice with a few standards. Taking advantage of the massive oak regeneration in the 1880s, the forest developed progressively into a high forest and was then invaded by beech. In 2001, the area was covered by even-aged oak trees and heterogeneously sized beech trees. In addition, an understory of hornbeam (*Carpinus betulus* L.) occurred in oak dominated areas. A 1 ha plot was installed in an intimate mixture of oak and beech, in

which all trees with a circumference higher than 15 cm were mapped (coordinates) and measured (stem circumference at a height of 1.3 m, total tree height, height of largest crown extension, height to crown base, crown diameters in two directions) at the end of the years 2001 and 2011.

Meteorological data were monitored with an automatic meteorological station located in an open field 300 m away from the forest site. Soil horizon properties were characterized based on the soil profile description and the measurements carried out

by Jonard et al. (2011).

To run the simulations, the values of some model parameters were taken directly from the literature. Other parameters involved in empirical relationships were fitted either with data from previous studies or with unpublished monitoring data collected in the study site or in the ICP Forests level II plots of Wallonia (Table 2). Potential explanatory variables of Eq. 31 used to estimate height growth were selected by applying a stepwise forward selection procedure based on the Bayesian Information

Criterion (BIC). A multivariate model was then adjusted with the selected variables (Appendix E). The parameters of the *npp* to *gpp* ratio relationship, the maintenance respiration per g of N at 15°C and the *PAR* use efficiency of sunlit and shaded leaves were adjusted with the nlm function of R (R Core Team, 2013) based on observed basal area increments (*BAIs*) using the maximum likelihood approach.

**2.4 Statistical evaluation of model predictions**

The quality of the model was evaluated for various combinations of model options (i.e., photosynthesis model of CASTANEA vs *PUE*, *npp* to *gpp* ratio *vs* temperature-dependent maintenance respiration, distance-dependent vs -independent crown extension), by comparing predicted and observed BAIs using several statistical indices and tests such as the normalized average error, the *P* value of the paired *t*-test, the regression test, the root mean square error, the Pearson's correlation and the modelling

efficiency (Janssens and Heuberger, 1995). For the regression test, the Deming fitting procedure (mcreg function of the mcr package in R) was retained to account for the errors on both the observations and the predictions.

The model quality was also evaluated based on its ability to reconstruct the size - growth relationships for sessile oak and European beech in the mixed stand of Baileux. The observed and predicted *BAIs* of the trees (calculated for the 2001 – 2011 period) were related to their girth at the beginning of the assessment period. A segmented regression was then applied to

observations and predictions to determine the girth threshold under which trees were not growing and to estimate the slope of the linear relationship between *BAI* and initial girth. The heteroscedasticity of the residuals was accounted by modelling their standard deviation with a power function of the initial girth. The fitting was carried out using the nlm function in R.





### 2.5 Simulation experiment

To illustrate how the model can be used to predict climate change impacts on forest ecosystem functioning, the growth dynamics in the mixed stand of Baileux was simulated according to three IPCC climate scenarios using the following options: photosynthesis model, *npp* to *gpp* ratio and distance-independent crown extension. The climate scenarios retained for this
study were obtained from the global circulation model CNRM-CM5 (Voldoire et al., 2013) based on the Representative Concentration Pathways for atmospheric greenhouse gases described in the Fifth Assessment Report of the Intergovernmental Panel on Climate Change (Collin et al., 2013). The Representative Concentration Pathways (RCP2.6, RCP4.5 RCP8.5) are characterized by the radiative forcing in the year 2100 relative to preindustrial levels (+2.6 W m$^{-2}$, +4.5 W m$^{-2}$, +8.5 W m$^{-2}$). The CNRM-CM5 describes the earth system climate using variables such as air temperature and precipitations on a low-
resolution grid (1.4° in latitude and longitude). Although reliable for estimating global warming, such a model fails to capture the local climate variations. Therefore, these climate projections were downscaled by the Royal Meteorological Institute of Belgium (RMI), using the regional climate model ALARO-0 (Giot et al., 2016). The meteorological files that were received from RMI are hourly values of the longwave and shortwave radiations, air temperature, surface temperature, rainfall, specific humidity, zonal and meridional wind speeds and atmospheric pressure with a 4 km spatial resolution. Specific humidity was
converted into relative humidity using the Tetens formula (Tetens, 1930). For a reference period (1976 - 2005), we compared the models predictions with observed meteorological data and detected some biases, especially for precipitations (overestimation of 27%). The biases were corrected by adding to the predictions (or by multiplying them with) a correction factor specific to the month (Maraun and Widmann, 2018). An additive correction factor was used for the bounded variables (radiations, precipitation, relative humidity, wind speed) and a multiplicative one for the other variables (air and surface
temperatures).

For the simulations, two 24-year periods (100 years apart) were considered. The period from 1976 to 1999 served as a historical reference while the rest of the simulations based on climate projections were conducted for the 2076-2099 period. The simulations were performed either by keeping the $CO_2$ concentration of the atmosphere constant (i.e., 380 ppm) or by allowing it to vary according to the years and climate scenarios. Each simulation started with the same initial stand (mixed stand of
Baileux in 2001) and lasted 24 years; a thinning operation (25% in basal area) was achieved in 1978 or 2078 and in 1990 or 2090 (12-year cutting cycle). The mean basal area increment obtained with the various climate scenarios were compared using the Tukey multiple comparison test.



## 3. Results

### 3.1 Reconstructed *npp* vs predicted *gpp*

Based on two successive stand inventories (2001 and 2011) and using HETEROFOR in reverse mode (see sect. 2.2.7), the individual *npp* was reconstructed and related to the *gpp* predicted with the photosynthesis method of CASTANEA. The linear
relationship between *npp* and *gpp* explained 81 and 83 % of the variability for sessile oak and for European beech, respectively (Fig. 2). The intercept was positive and significantly different from 0 but did not differ between the two trees species. The slope of the relationship was higher for sessile oak (0.54) than for European beech (0.40).

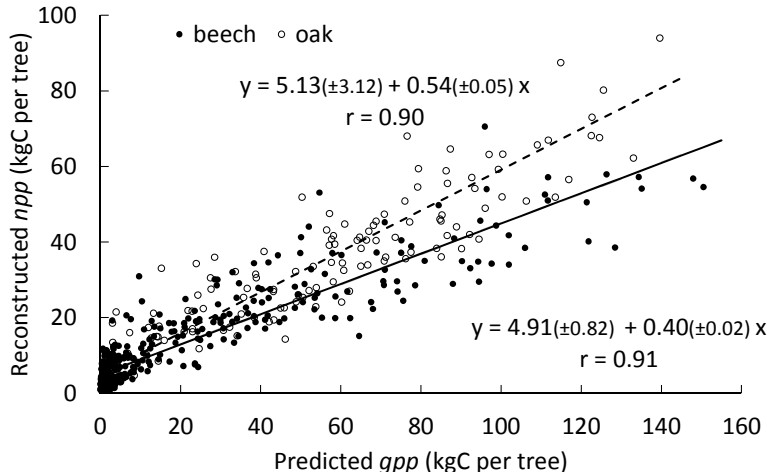

**Figure 2. Relationship between the individual *npp* reconstructed based on successive stand inventories (2001 and 2011) and the *gpp* predicted with the process-based option (photosynthesis method of CASTANEA). Values in parentheses are 95% confidence intervals for the intercept and the slope in the equations. The Pearson's correlation between *npp* and *gpp* is indicated on the graph.**

### 3.2 Model performance in predicting individual basal area increment (*BAI*)

HETEROFOR was run with height different combinations of options for describing photosynthesis (biochemical model of
CASTANEA *vs PUE*), respiration (*npp* to *gpp* ratio *vs* temperature-dependent maintenance respiration) and crown extension (distance-dependent *vs* -independent). The mean observed and predicted *BAI*s were not significantly different from each other (Paired *t*-test), except for European beech with the *PUE* approach and for sessile oak for one combination of options: *PUE/npp* to *gpp* ratio/distance-dependent crown extension (Table 3). To go further, the observed *BAI*s were regressed on the predicted *BAI*s using the Deming regression and the 95% confidence intervals of the intercept and the slope were calculated to evaluate
if the regression line significantly differed from the 1:1 line. For the simulations using the CASTANEA photosynthesis, the

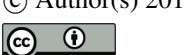


intercept was closer to 0 and the slope closer to 1 when HETEROFOR was run with the *npp* to *gpp* ratio approach compared with the temperature-dependent maintenance respiration. This difference between respiration options was not observed for the *PUE* approach (Table 3). For the simulations using the CASTANEA photosynthesis, the smaller RMSE, the higher Pearson's correlation and modelling efficiency were obtained with the *npp* to *gpp* ratio. The distance-independent crown extension

5 provided slightly more accurate results than the distance-dependent one for European beech while the reverse was true for sessile oak. For the *PUE* approach, the best combination of options was the *npp* to *gpp* ratio and the distance-independent crown extension (Table 3). In summary, the biochemical model of CASTANEA and the *npp* to *gpp* ratio approach provided better predictions than the alternative options and the way crown extension was described had little impacts on the prediction quality (Table 3, Fig. 3).

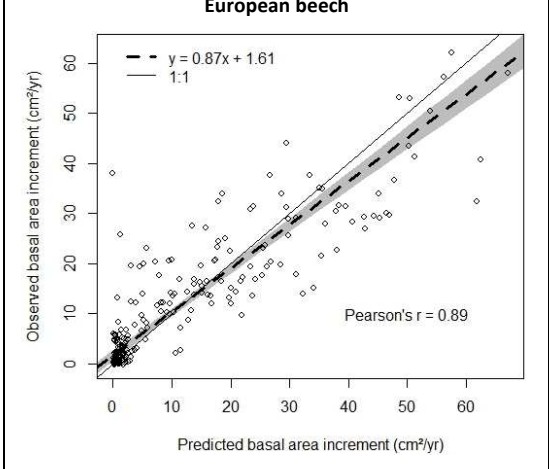

**Figure 3. Comparison of observed and predicted basal area increments (*BAIs*) for the simulation with the photosynthesis method of CASTANEA, the *npp* to *gpp* ratio approach to account for tree respiration and the distance-independent crown extension (see Table 3). The dashed line represents the Deming regression between observations and predictions with the shaded area indicating the 95%**
15 **confidence interval and the solid line the 1:1 relationship.**

### 3.3 Reconstructing size – growth relationships

The size - growth relationships were very similar between observations and predictions, except for European beech with the *PUE* approach (Fig. 4). In this case, the reconstructed size-growth relationship underestimated the observed one (Fig. 4). The proportion of the *BAI* variance explained by the size - growth relationship ($R^2$) was higher for European beech than for sessile

20 oak for both observations and predictions, for observations than for predictions (especially regarding sessile oak) and when simulations were carried out with the CASTANEA option rather than with the *PUE* approach (Fig. 4). Regarding the



observations, the girth threshold was lower for European beech (50.6 cm) than for sessile oak (74.5 cm) while the slopes of the relationship were similar (Fig. 4).

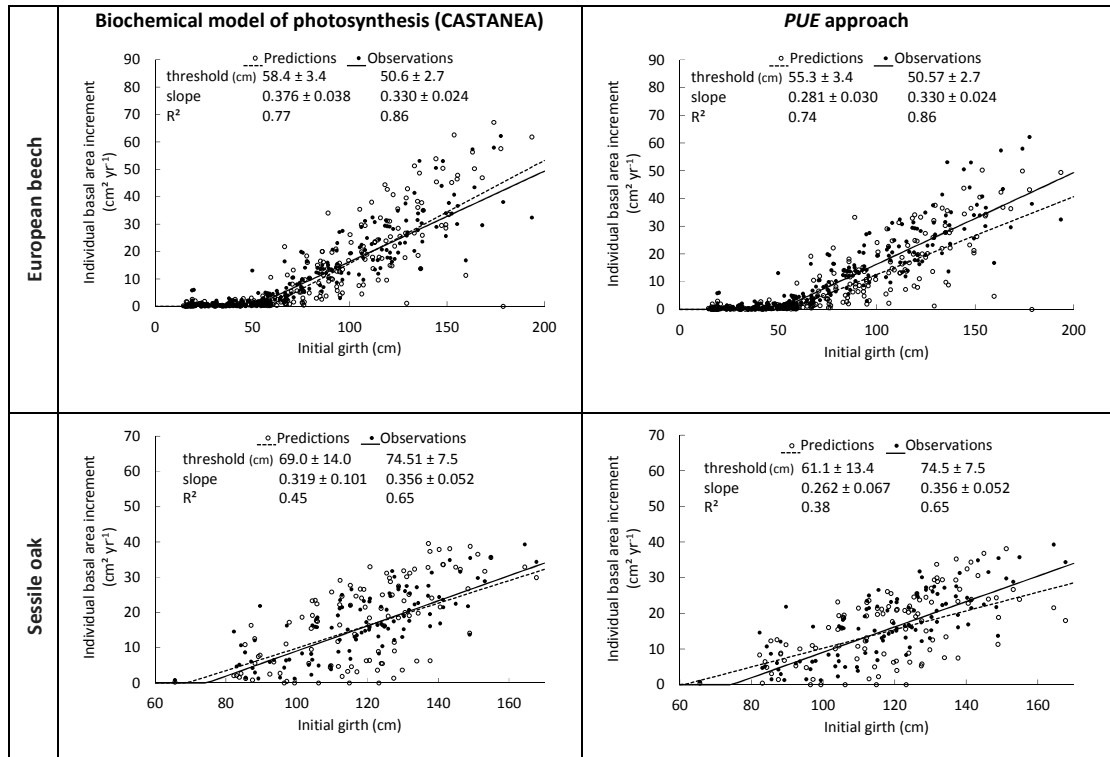

**Figure 4. Reconstruction of the size - growth relationships for sessile oak and European beech using the photosynthesis method of CASTANEA (left panel) or the *PUE* approach (right panel), the *npp* to *gpp* ratio approach to account for tree respiration and the distance-independent crown extension. The predicted relationships between the individual *BAI* (calculated for the 2001–2011 period) and the initial girth are compared with observed ones. The solid and dashed lines represent the segmented regression applied respectively to observations and predictions to determine the girth threshold under which trees were not growing and to estimate**
**the slope of the linear relationship between *BAI* and initial girth. The 95% confidence intervals for the intercept and the slope are provided as well as the R² of the model.**

### 3.4 Simulation of climate change impact on tree growth

When the $CO_2$ concentration of the atmosphere was fixed, no effect of the climate scenario was detected on stand *BAI* but a

slight impact was assessed on sessile oak *BAI* which was higher for the RCP2.6 than for the historical scenario (Fig. 5). For



the simulations with a variable atmospheric $CO_2$ concentration, the difference in total, sessile oak and European beech *BAI* were much more pronounced between climate scenarios. For the whole stand as well as for oak and beech separately, *BAI* increased in the order - historical, RCP2.6, RCP4.5 and RCP8.5 -, with the stand *BAI* of these RCP scenarios being between 17 and 72% higher than that of the historical scenario. All scenarios had a *BAI* significantly different from each other, except

RCP2.6 and RCP4.5 for the whole stand and the two tree species and historical and RCP2.6 for European beech (Fig. 5).

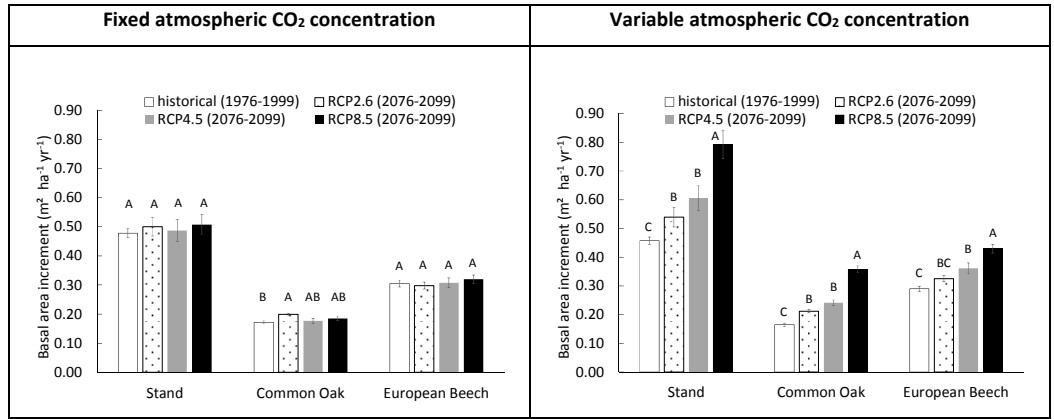

**Figure 5. Basal area increment (*BAI*) of the mixed stand in Baileux (and of its two main tree species) simulated with climate scenarios produced with the GCM model CNRM-CM5, downscaled with ALARO-0 and corrected empirically for remaining biases. The**
**simulations were performed by using the Castanea method to calculate photosynthesis, the *npp* to *gpp* ratio approach and a distance-independent description of crown extension. The $CO_2$ concentration of the atmosphere was either kept constant (left) or increased with time according to the climate scenario considered (right). Two time periods were considered. 1976-1999 was used as a reference period for running the model with the historical climate scenario while the simulations with future climate scenarios were achieved for the 2076-2099 period. The climate scenarios were based on the representative concentration pathways for atmospheric**
**greenhouse gases described in the fifth assessment report of IPPC. For a given tree species and $CO_2$ concentration modality, the scenarios with common letters have a *BAI* not significantly different from each other (α=0.05).**



## 4. Discussion

Very few tree-level, process-based and spatially explicit models have been developed and these often contain only some of the modules necessary to estimate resource availability (solar radiation, water and nutrients). While a description of these models is generally available in the literature, their evaluation by comparison with tree growth measurements is not always accessible

or was carried out based on stand-level variables. We have therefore very few information to compare the performances of HETEROFOR at the tree level with those of similar models. Simioni et al. (2016) faced the same problem with the NOTG 3D model.

HETEROFOR first estimates the key phenological dates, the radiation interception by trees and the hourly water balance (de Wergifosse et al., in prep). Then, based on the absorbed *PAR* radiation, individual *gpp* is calculated with a *PUE* approach or

with the photosynthesis routine of CASTANEA (Dufrêne et al., 2005). When selecting the *npp* to *gpp* ratio (the most accurate option to account for tree respiration), the photosynthesis routine of CASTANEA provided better predictions than the *PUE* efficiency for the distance-dependent crown extension and both approaches performed similarly for the distance-independent crown extension (Table 3). This is quite encouraging that the process-based approach for estimating photosynthesis provided predictions of the same -or even better- quality than the empirical approach fitted with tree growth data taken on the study site.

If no extrapolation to future climate is required, the *PUE* approach remains however still valuable, especially when meteorological data are lacking. For the mixed stand in Baileux, we related the *npp* reconstructed from successive tree inventories with the *gpp* predicted based on the CASTANEA approach (Fig. 2). The good linear relationships (Pearson's correlation > 0.90) obtained for both oak and beech make us confident in the adaptation of the photosynthesis routine of CASTANEA for the tree level. Furthermore, since the parameters of the photosynthesis routine were taken directly from

CASTANEA and not calibrated specifically for HETEROFOR, one can expect that the agreement between the predicted *gpp* and the reconstructed *npp* could still be improved.

When comparing the two options available in HETEROFOR for converting *gpp* into *npp*, model performances are generally better with the *npp* to *gpp* ratio approach than with the temperature-dependent routine for maintenance respiration calculation (Table 3), except for sessile oak with the *PUE* approach. This can be explained since the error in the maintenance respiration

calculation results from various sources. At the tree compartment level, uncertainties in the estimation of biomass, sapwood proportion, nitrogen concentration and temperature are multiplied (Eq. 11). Then, the errors made on all tree compartments are summed up. Among these uncertainty sources, the inaccuracy in the estimation of the sapwood proportion could explain why the maintenance respiration routine provided better results for sessile oak than for European beech (Table 3). Since the sapwood of sessile oak can easily be distinguish from the heartwood based on the colour change, we had a lot of sapwood

measurements available to fit a relationship. For European beech, this was not the case; instead, we used a sapwood relationship obtained based on sap flow measurements (Jonard et al., 2011). This relationship could certainly be improved by direct measurements of sapwood made after staining the wood to highlight the living parenchyma. Another way to improve these relationships is to consider the social status of the trees since dominant trees have a higher sapwood depth than the suppressed



one (Rodriguez-Calcerrada et al., 2015). We tried to account for this by estimating the sapwood area based on the tree growth rate but it did not significantly increased the quality of the predictions. The poor performances obtained with the maintenance respiration option also indicates that the processes at play are still poorly understood and that further research are needed on this topic.

The process-based approach for estimating maintenance respiration accounts explicitly for the temperature effect through a $Q_{10}$ function. With the *npp* to *gpp* ratio approach, temperature is considered more indirectly by assuming that it affects respiration and photosynthesis in the same proportion, which is valid only in a given range of temperature (<20°C) and for non-stressing conditions. Indeed, the optimum temperature for photosynthesis is between 20 and 30°C while the optimum temperature for respiration is just below the temperature of enzyme inactivation (>45 °C). Therefore, between 30 and 45°C,

photosynthetic rates decrease, but respiration rate continues to increase (Yamori et al., 2013). In addition, while water stress reduces both photosynthesis and respiration, its effect on the two processes is not necessarily equivalent (Rodriguez-Calcerrada et al., 2014). Compared to the *npp* to *gpp* ratio approach, the maintenance respiration calculation seems more appropriate to simulate the impact of climate change but this is at the expense of less accurate predictions at the tree level. The ideal is to compare the two options to evaluate the prediction uncertainty associated with the modelling of respiration. Alternatively, one

can choose one or the other option depending on the aim of the simulation.

Beside these contrasted model performances between the two options used to assess respiration, differences were also observed between both options adopted to model crown extension, with slightly better predictions when using the distance-independent approach compared to the distance-dependent one. However, we should not put aside the distance-dependent approach based on this first comparison. The differences in prediction quality between the two methods were quite small, probably because

the length of the simulation was not sufficient to drastically affect the crown dimensions which had been initialized based on measurements. In addition, describing mechanisms that governs crown development in interaction with neighbours (mechanical abrasion, crown interpenetration) is indeed crucial to capture non-additive effect of species mixtures (Pretzsch, 2014). By accounting for crown plasticity, our distance-dependent approach could help better understand how uneven-aged and mixed stands optimize light interception by canopy packing and how they increase productivity (Forester and Albrecht,

2014; Juncker et al., 2015). To fully evaluate the interest of this approach, the predicted crown development should be compared with precise crown measurements repeated over several decades and taken in a large diversity of stand structures.

Based on the current evaluation, the process-based variant perform better than the more empirical one for photosynthesis but not for respiration and crown extension, probably because the processes are better known for photosynthesis. This option combination had a high modelling efficiency, especially regarding European beech (Table 3). The Pearson's correlation

between measurements and predictions of individual basal increment amounted to 0.89 and 0.69 for European beech and sessile oak, respectively. By comparison, Grote and Pretzsch (2002) obtained a correlation of 0.60 for the individual volume of beech trees with the BALANCE model. This lower correlation can partly be explained by the integration of the uncertainty on tree height in the volume estimations. The HETEROFOR performances in terms of tree growth are quite good and could still be improved by a Bayesian calibration of the parameters.





Individual *npp* and retranslocated C are allocated first to foliage and fine roots and then partitioned between above- and below-ground structural compartments. Based on the derivative and rearrangement of a biomass allometric equation, the increment in aboveground structural biomass is used to estimate the combined increment in *dbh* and height. This results in a system of one equation with two unknowns (increment in *dbh* and height). We decided to resolve it by fixing the height growth based on

a relationship taking into account tree size (*dbh* or height), the height growth potential (height increment if all the remaining carbon was allocated to height growth) and a light competition index. An intermediate level of sophistication was adopted to describe height growth, between the simple height-*dbh* allometry and the fine description of tree architecture of functional-structural models. Height-*dbh* relationships provides a static picture in which age and neighbour effects are confounded and are not suitable to describe individual growth trajectories (Henry and Aarsen, 1999). More sophisticated relationships

considering age and dominant height can be used for even-aged stands (Le Moguédec and Dhôte, 2012) but are hardly applicable in uneven-aged stands for which tree age is unknown. On the other hand, the functional-structural models based on resource availability at organ level and using a short time step can only be applied to a limited number of trees given the long computing times (Letort et al., 2008).

Our individual height growth model was fitted with height data measured ten years apart (Appendix E). A large uncertainty

was however associated to these data. First, height measurements were obtained to the nearest meter given the difficulty to clearly identify the top of the trees in closed canopy forests. Second, two errors were added since the height increment was obtained by the difference between two height measurements. Consequently, the uncertainty was more or less of the same order of magnitude than the expected height growth in ten years. Despite these uncertainties, a substantial part of the variability was explained by the model (72% for European beech, 43% for oak). Among the variables tested, the height growth potential

had the main effect, which is not surprising since this height growth potential noisily contains the information on height increment. We were also able to highlight the effect of light competition. For a same height growth potential, trees undergoing stronger light competition seem to invest more carbon for height growth than for *dbh* increment (Fig. E1 in Appendix E), which is corroborated by results of other studies (e.g., Lines et al., 2012). This strategy aims at minimizing overtopping by neighbours and maximizing light interception (Jucker et al., 2015). Trouvé et al. (2015) found similar results and showed the

positive effect of stand density on height growth in the allocation between height and diameter increment in even-aged stands of sessile oak. The decrease in the red:far red ratio of incident light promotes apical dominance and internode elongation through the phytochrome system (shade avoidance reaction, Henry and Aarsen, 1999).

We were also quite satisfied to observe that the model was able to reproduce the size-growth relationship. This relationship is characterized by two parameters: the threshold which defines the minimum girth for radial growth to occur and the slope

providing the theoretical maximum growth efficiency (Le Moguédec and Dhôte, 2012). For European beech, the observed threshold was 49.1 cm and was easy to detect visually since there were many trees with a girth inferior to that exhibiting nearly no basal area increment. For sessile oak, the observed threshold was higher than for European beech (70.8 cm) and nearly all trees had a higher girth. This can be related to the fact that sessile oak is a less shade-tolerant species than European beech. The observed slope was however the same for both tree species meaning that they have the same maximum growth efficiency.





For the size-growth relationship reconstructed with HETEROFOR, the predicted threshold and slope were generally not significantly different from the observed ones but the predicted size-growth relationships explained a lower proportion of the variability, especially for sessile oak.

To illustrate one possible application of HETEROFOR, a simulation experiment was achieved and allowed us to compare the radial growth predicted for 2076-2099 according to three IPCC scenarios with that simulated for an historical period (1976-1999). When atmospheric $CO_2$ concentration was kept constant (380 ppm), differences among scenarios remained non-significant, except for sessile oak displaying a slightly higher basal area increment for the RCP2.6 than for the historical scenario (Fig. 5). Analyzing in-depth the model outputs, we found that this lack of effects resulted from a balance between negative and positive impacts of climate change. While the increase in air temperature (+0.9 and 3.7°C for RCP2.6 and 8.5) and in the vegetation period length (+8 and 37 days for RCP2.6 and 8.5) favoured photosynthesis, the more frequent and intense water stress negatively affected it (data not shown). The positive and negative effects of climate change were of the same magnitude for both tree species and offset each other. For the simulations with a variable atmospheric $CO_2$ concentration, the differences among scenarios were much larger highlighting a strong $CO_2$ fertilization effect for both sessile oak and European beech (Fig. 5). These results are in agreement with Reyer et al. (2014) who used the 4C model to predict productivity change in Europe according to a large range of climate change projections. They found NPP increases in most European regions (except a few cases in Mediterranean mountains) when considering persistent $CO_2$ effects by using variable atmospheric $CO_2$ concentration. Assuming an acclimation of photosynthesis to $CO_2$ (by maintaining atmospheric $CO_2$ constant), they predicted increases in Northern, decreases in Southern and ambivalent responses elsewhere in Europe. Similar response patterns were also obtained by Morales et al. (2007). Rötzer et al. (2013) used the BALANCE model to compare the impact of future and current climate conditions on the productivity of beech in Germany and showed a 30% decrease in NPP without considering the rise in atmospheric $CO_2$ concentration. After evaluating CASTANEA against eddy covariance and tree growth data in a few highly instrumented sites, Davi et al. (2006) simulated the trend in GGP and net ecosystem productivity (NEP) in these sites from 1960 to 2100. For sessile oak and European beech, they obtained a 53% and 67% increase in GPP and NEP, respectively.

Given the magnitude of the $CO_2$ fertilization effect (leading to a 72% increase in basal area increment in 100 years for RCP8.5), we conducted retrospective simulations to check that HETEROFOR reproduces well the increase in productivity observed by Bontemps et al. (2011) for beech forests in the north-east of France (data not shown). Based on historical atmospheric $CO_2$ concentrations, we simulated radial growth during two periods (1879-1910 *vs* 1979-2010) using the same climate data (obtained by re-analysis for 1979-2010). These simulations showed a productivity increase of 12% over 100 years. By comparison, Bontemps et al. (2011) reported productivity increases ranging from 10 to 70% over 100 years depending on the nitrogen status of the forest. The increase in radial growth simulated with HETEROFOR for the mixed stand in Baileux (Fig. 5) seems therefore plausible but assumes unchanged nutritional status. Increased productivity generates however higher nutrient demand by trees, which is not systematically satisfied by larger soil nutrient supply, especially in the poorest sites. Consequently, the augmentation of forest productivity will most likely be constrained by nutrient availability and give rise to





a deterioration of the nutritional status as already observed across Europe (Jonard et al., 2015). To improve our predictions, nutritional constraints must be taken into account. In this perspective, a mineral nutrition and nutrient cycle module was incorporated in HETEROFOR. As it was developed in parallel to the water balance, some adaptations are needed for a perfect coupling of the two modules (e.g., change from an annual to a monthly time step for soil chemistry update). A complete

5 description and evaluation of the nutrient module will be provided in a future study.



### 5. Conclusion and future prospects

Our ambition was to develop a model responsive to both management actions and environmental changes that would be particularly well adapted to mixed and uneven-aged stands. We thought that this model had to be tree-level, spatially explicit and process-based. Except BALANCE and more recently NTOG 3D, no such a model existed in the literature. To fill this gap,

we elaborated the HETEROFOR model based on concepts quite different from those used for BALANCE. In this study, a first evaluation of the model performances showed that HETEROFOR predicts well individual radial growth and is able to reproduce size-growth relationships. We also noticed that the more empirical options for describing maintenance respiration and crown extension provide the best results while the process-based approach best performs for photosynthesis.

Here, only the core of HETEROFOR was described. The water balance and phenology modules are presented and evaluated

in a companion paper (de Wergifosse et al., in prep) while the radiation transfer and nutrient modules will be described later. For the next steps, we plan to couple HETEROFOR with existing libraries such as regeneration, genetics and economics. As HETEROFOR was developed within the CAPSIS platform, it is continually improving thanks to the collaborative dynamics among modellers.

A broader assessment of the model performances will be carried out based on forest monitoring plots distributed all over

Europe. Indeed, HETEROFOR was designed to be particularly suitable for the level II plots of ICP Forests. The processes were described at a scale that facilitates the comparison between model predictions and observations. Many data collected in these plots can be used to initialize and run the model or to calibrate and evaluate it. HETEROFOR can also be seen as a tool for integrating forest monitoring data and quantifying non-measured processes. While it is now calibrated for oak and beech forests, HETEROFOR will be parameterised for a large range of tree species in order to use it for testing and reproducing

identity and diversity effects.

Given all the uncertainties related to climate change impacts, it is an illusion to believe that a model will predict accurately the future dynamics of forest growth. However, models such HETEROFOR can be very useful to compare scenarios. Among others, HETEROFOR can be used to select the management options that maximise ecosystem resilience or to quantify uncertainty in the response of forest ecosystem to climate change.



## 6. Code availability

The source code of CAPSIS and HETEROFOR is accessible to all the members of the CAPSIS co-development community. Those who want to join this community are welcome but must contact François de Coligny (coligny@cirad.fr) or Nicolas Beudez (nicolas.beudez@inra.fr) and sign the CAPSIS charter (http://capsis.cirad.fr/capsis/charter). This charter grants access

on all the models to the modellers of the CAPSIS community but only to them. The modellers may distribute the CAPSIS platform with their own model but not with the models of the others without their agreement. CAPSIS4 is a free software (LGPL licence) which includes the kernel, the generic pilots, the extensions and the libraries. HETEROFOR is basically not free and belongs to its authors who decided to distribute it through an installer containing the CAPSIS4 kernel and the latest version (or any previous one) of HETEROFOR upon request from Mathieu Jonard (mathieu.jonard@uclouvain.be). The

version 1.0 used for this paper is temporarily available at http://amap-dev.cirad.fr/projects/capsis/files. The end-users can install CAPSIS from an installer containing only the HETEROFOR model while the modellers who signed the CAPSIS charter can access to the complete version of CAPSIS with all the models. Depending on your status (end-user vs modeller or developer), the instructions to install CAPSIS are given on the CAPSIS website (http://capsis.cirad.fr/capsis/documentation). The source code for the modules published in Geoscientific Model Development (Jonard *et al.,* submitted; de Wergifosse *et*

*al.*, in prep.) can be downloaded from https://github.com/jonard76/HETEROFOR-1.0 (DOI 10.5281/zenodo.3242014).



## 7. Data availability

The data used in this paper are available through the input files for HETEROFOR which are embedded in the installer (see sect. 6).





## 8. Appendices

### 8.1 Appendix A – Description of the soil heat transfer routine

The temperature of the mineral soil (T in °C) is calculated by soil depth increment ($\Delta z$ in m) using a simplification of the soil heat transfer equation assuming a constant thermal diffusivity (D in m² s⁻¹) across the soil profile. The thermal diffusivity
characterizes the rate of heat transfer within the soil and corresponds to the ratio of the thermal conductivity ($K$ in W m⁻¹ K⁻¹) to the volumetric heat capacity ($c_v$ in J m⁻³ K⁻¹).

$$\frac{\partial T}{\partial t} = \frac{1}{c_v} \cdot \frac{\partial}{\partial z} \cdot \left(K \frac{\partial T}{\partial z}\right) => \frac{\partial T}{\partial t} = D \cdot \frac{\partial^2 T}{\partial z^2} \tag{50}$$

Eq. (50) can be rewritten as follows according to Anlauf and Liu (1990) and Baker and Don Scott (1998):

$$T_{z,t+\Delta t} = T_{z,t} + D \cdot \frac{\Delta t}{\Delta z^2} \cdot \left(T_{z+\Delta z,t} + T_{z-\Delta z,t} - 2T_{z,t}\right) \tag{51}$$

The soil depth increment can be chosen by the user but it has to be smaller than one third of the thiniest horizon. The soil depth increment can be slightly modified by the model to ensure the soil depth is a multiple of the soil depth increment. Then, a stability criterion is checked for each hour and if it is not respected, the temporal step is divided by two.

$$K \cdot \frac{\Delta T}{\Delta z^2} < 0.5 \tag{52}$$

The thermal diffusivity is calculated for each soil horizon based on the thermal conductivity and the volumetric heat capacity and then averaged by weighing according the horizon thickness. The thermal conductivity is obtained with the empirical model of Kersten (1949):

$$K = 0.1442 \cdot (0.9 \cdot \log(\vartheta) - 0.2) \cdot 10^{0.6243\rho_b} \text{ (for silt or clay soils)} \tag{53}$$

$$K = 0.1442 \cdot (0.7 \cdot \log(\vartheta) + 0.4) \cdot 10^{0.6243\rho_b} \text{ (for sandy soils)} \tag{54}$$

with   $\vartheta$, the gravimetric soil water content (g g⁻¹),

$\rho_b$, the bulk density (kg m⁻³).

The volumetric heat capacity of soils is approximated through a separation of the soil constituents in solid and liquid phases:

$$c_v \simeq 836 \cdot \rho_b + 4180 \cdot \vartheta \cdot \rho_b \cdot 1000 \cdot \rho_w \tag{55}$$

with   $\rho_w$, the volumetric mass of water (kg m⁻³).

To initialize the procedure, the top and bottom temperature during the whole simulation and the initial temperature at each soil depth must be known. The soil temperature at the top of the mineral soil (just under the forest floor) is given by Eq. (56) adapted from van Wijk and de Vries (1963) and Cichota et al. (2004). The bottom temperature is fixed and corresponds to the mean annual air temperature. This assumption can be made as the soil depth largely exceeds 1 meter. The initial temperature
is found through a simple interpolation of the temperatures between the soil interface and the bottom.

$$T_t = \overline{T}_y + \frac{(\overline{T}_{d-1} - \overline{T}_y)}{A_{air}} \cdot A_{soil} + \frac{a_{air}}{2} \cdot red_d \cdot \sin\left(\omega\left(t - t_{T_{max}}\right) + \frac{\pi}{2} - \omega\frac{\Delta z}{Damping}\right) \tag{56}$$



with $\overline{T}_y$, mean annual air temperature (°C),

$\overline{T}_{d-1}$, mean air temperature of the previous day (°C),

$A_{air}$, annual air temperature amplitude corresponding to the difference between the maximum and the minimum mean daily temperature over the year (°C),

$A_{soil}$, parameter corresponding to the mean annual soil temperature amplitude (°C),

$a_{air}$, daily air temperature amplitude ($T_{max} - T_{min}$) calculated over the 24 hour period centered on the considered time (°C),

$red_d$, parmeter reducing the daily air temperature amplitude to the daily soil temperature amplitude (fixed to 0.13)

$\omega$, radial frequency (h$^{-1}$) $= \frac{2\pi}{24}$,

$t_{T_{max}}$, hour of the day at which air temperature is maximal (as the sinusoidal shape of the diurnal soil temperature cycle is not perfectly symmetric, $t_{T_{max}}$ is adapted so that the period between maximum and minimum soil temperature is exactly 12 hours),

$\Delta z$, thickness of organic horizons (m),

Damping, parameter accounting for the phase shift between the diurnal cycle of the air and soil temperature (fixed to

0.0853 after calibration).

The temperature of the organic horizons was obtained as the mean between air temperature and the temperature at the interface between organic horizons and mineral soil.



### 8.2 Appendix B – Development of Eq. (29)

Equation (29) can be developed in order to isolate $\Delta h$:

$$\Delta\left(dbh^2 \cdot h\right) = (dbh + \Delta dbh)^2 \cdot (h + \Delta h) - dbh^2 \cdot h \tag{57}$$

$$\Delta\left(dbh^2 \cdot h\right) = (dbh^2 + 2 \cdot dbh \cdot \Delta dbh + (\Delta dbh)^2) \cdot (h + \Delta h) - dbh^2 \cdot h$$

$$\Delta\left(dbh^2 \cdot h\right) = dbh^2 \cdot h + 2 \cdot dbh \cdot \Delta dbh \cdot h + (\Delta dbh)^2 \cdot h + dbh^2 \cdot \Delta h + 2 \cdot dbh \cdot \Delta dbh \cdot \Delta h + (\Delta dbh)^2 \cdot \Delta h - dbh^2 \cdot$$

$$h\Delta\left(dbh^2 \cdot h\right) = \Delta dbh^2 \cdot (h + \Delta h) + (\Delta dbh)^2 \cdot (h + \Delta h) + dbh^2 \cdot \Delta h$$

$$\Delta\left(dbh^2 \cdot h\right) = \Delta h \cdot (\Delta dbh^2 + (\Delta dbh)^2 + dbh^2) + h \cdot (\Delta dbh^2 + (\Delta dbh)^2) \tag{58}$$

Considering $(\Delta dbh)^2 \ll \Delta dbh^2 \ll dbh$, the following approximation can be done:

$$\Delta\left(dbh^2 \cdot h\right) \cong \Delta h \cdot dbh^2 + h \cdot \Delta dbh^2 \tag{59}$$

$$\Delta h \cdot dbh^2 \cong \Delta\left(dbh^2 \cdot h\right) - h \cdot \Delta dbh^2 \tag{60}$$

$$\Delta h \cong \frac{\Delta\left(dbh^2 \cdot h\right)}{dbh^2} - \frac{h \cdot \Delta dbh^2}{dbh^2} \tag{61}$$





### 8.3 Appendix C - Delevoy height estimation

The Delevoy height is the height at which stem diameter is half the diameter at breast height and is calculated as follows from taper (cm m$^{-1}$):

$$h_{del} = 1.3 + \frac{dbh - \frac{dbh}{2}}{taper} \tag{62}$$

where the taper is obtained based on the girth at 10% of the tree height (G10%) and the relative girth at 60% of the tree height (RG60%) for which empirical equations are provided by Dagnelie *et al.* (1999) for several temperate tree species:

$$taper = \frac{(1 - CR60\%) \cdot C10\%}{0.5 \cdot h \cdot \pi} \tag{63}$$

10      with

$$C10\% = a + b \cdot \pi \cdot dbh + c \cdot (\pi \cdot dbh)^2 + d \cdot (\pi \cdot dbh)^3 + e \cdot h + f \cdot (\pi \cdot dbh)^2 \cdot h \tag{64}$$

$$CR60\% = a + \frac{b}{C10\%} + \frac{c}{C10\%^2} \tag{65}$$

15   **Table C1. Parameters of Eqs. (64) and (65) for oak and beech according to Dagnelie et al. (1999)**

|        | a      | b      | c             | d            | e        | f             |
|--------|--------|--------|---------------|--------------|----------|---------------|
| **Oak** |        |        |               |              |          |               |
| C10%   | 3.9330 | 1.0284 | -0.31611 10$^{-3}$ | 0.44036 10$^{-6}$ | -0.33113 | -0.28051 10$^{-5}$ |
| CR60%  | 0.4838 | 14.667 | -405.67       |              |          |               |
| **Beech** |      |        |               |              |          |               |
| C10%   | 3.8541 | 1.0235 | -0.36276 10$^{-3}$ | 0.40063 10$^{-6}$ | -0.30551 | -0.20411 10$^{-5}$ |
| CR60%  | 0.5286 | 0      | 0             |              |          |               |

**8.4 Appendix D – Estimation of the height of largest crown extension (*hlce*) at equilibrium**

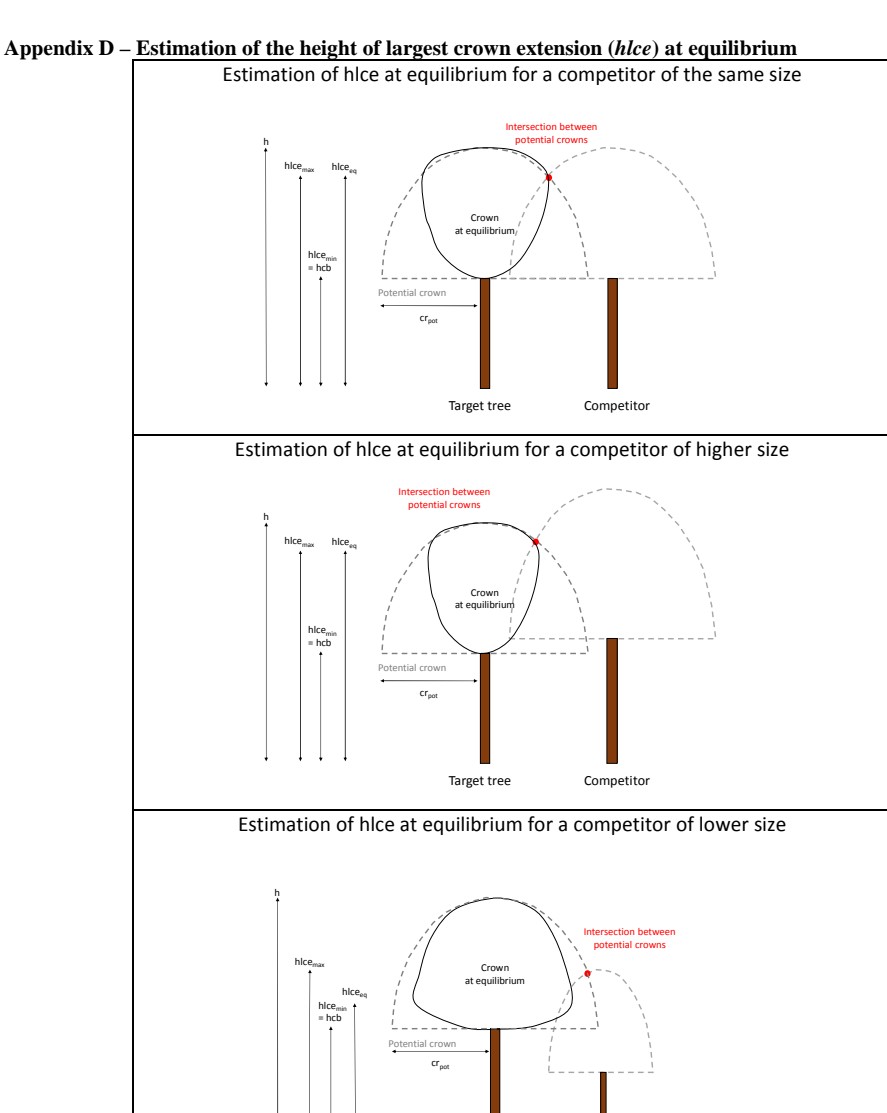

**Figure D1. Illustration of the routine used to determine the height of largest crown extension at equilibrium (*hlce_eq*) of a target tree in three contrasted situations of competition. A first step consists in determining the intersection between the potential crown of the target tree and the competitor. Then, the *hlce_eq* is fixed between the maximum *hlce* (corresponding to the intersection between potential crowns) and the minimum *hlce* (which the height to crown base) based on the relative height of the competitor.**





### 8.5 Appendix E – Height growth modelling results

The main factor explaining the height increment was the so-called height growth potential ($\Delta h_{pot}$) with a quadratic effect for beech and a cubic effect for oak (Table E1, Fig. E1). For both tree species, the light competition index (*LCI*) had a negative effect on height increment, meaning that, for a same height growth potential, trees under stronger competition for light had a

5   higher height growth than trees within better light conditions. For European beech, the variable selection procedure led to select height (which had a negative effect) to account for tree size while *dbh* was retained for sessile oak and had a positive effect. Even if the root mean square error was slightly higher for European beech (0.094) than for sessile oak (0.083), the height growth model explained a much larger proportion of the variability for European beech (72%) than for sessile oak (43%), partly because the height growth range was higher for European beech.

**Table E1. Parameters, R² and RMSE of the height growth model (Eq. 31) for European beech and sessile oak.**

|  | **European beech** | **Sessile oak** |
| --- | --- | --- |
| intercept | 0.0233 | -0.0562 |
| *dbh* (in cm) |  | 0.0023 |
| *h* (in m) | -0.0048 |  |
| *LCI* | -0.2556 | -0.1874 |
| $(\Delta dbh^2 h_{tot})/dbh^2$ (in m) | 0.6631 | 0.8183 |
| $[(\Delta dbh^2 h_{tot})/dbh^2]^2$ | -0.1777 | -0.9178 |
| $[(\Delta dbh^2 h_{tot})/dbh^2]^3$ |  | 0.4733 |
| RMSE | 0.09397 | 0.083017 |
| R² | 0.72 | 0.43 |



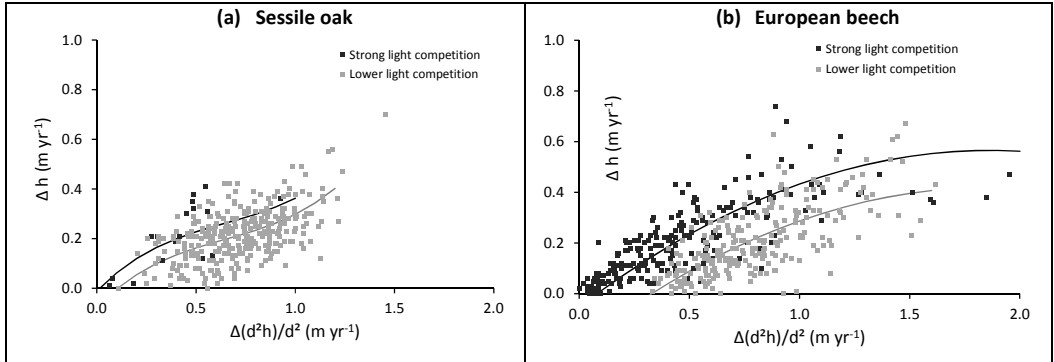

**Figure E1. Effect of the height growth potential on oak and beech height growth for two levels of light competition (strong light competition = light competition index ≤ 0.15, lower light competition = light competition index > 0.15). The solid lines represent the model predictions obtained using Eq. (31) with parameter values of Table E1 and with mean values for *dbh*, height or the light**
5 **competition index.**



**9. Author contribution**

MJ, FA, FdC, LdW, NB and HD developed the model code. FA carried out the calibration. MJ performed the simulations and analysed the model outputs. MJ prepared the manuscript with contributions from all co-authors.

**10. Competing interests**

5   The authors declare that they have no conflict of interest.

**11. Acknowledgements**

This work was supported by the *Service Public de Wallonie* (SPW/DGO 3/DNF) through the *Accord-Cadre de Recherche et Vulgarisation Forestières 2014–2019* and by the *Fonds de la Recherche Scientifique – FNRS* under the PDR-WISD Grant n°09 (project SustainFor) and the FRIA grant n°1.E005.18 (LdW PhD fellowship). We are also grateful to the RMI (Royal
10   Meteorological Institute of Belgium) for providing us climate projection data for the Baileux site.





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





**Table 1. Mixed stand characteristics per tree species derived from stand inventories in 2001 and 2011. Standard deviation is provided in parentheses.**

| Year | Tree species | Density (N ha⁻¹) | Basal area (m² ha⁻¹) | gbh[1] (cm) | height (m) | Crown dimensions radius (m) | length (m) |
|---|---|---|---|---|---|---|---|
| 2001 | | | | | | | |
| | hornbeam | 10 | 0.1 | 29.3 (18.2) | 8.9 (4.3) | | |
| | sessile oak | 121 | 13.3 | 115.5 (21.1) | 24.4 (1.7) | 3.6 (0.9) | 7.0 (1.7) |
| | European beech | 350 | 16.5 | 66.1 (39.3) | 18.2 (6.9) | 3.8 (1.0) | 8.9 (2.8) |
| 2011 | | | | | | | |
| | hornbeam | 6 | 0.0 | 22.6 (9.0) | 8.4 (2.7) | | |
| | sessile oak | 114 | 14.8 | 126.0 (21.9) | 26.9 (1.7) | 4.1 (2.5) | 7.7 (2.0) |
| | European beech | 260 | 17.7 | 80.2 (46.1) | 21.3 (7.3) | 4.2 (1.1) | 10.7 (3.3) |

[1] Girth at breast height



**Table 2. Description of model parameters for sessile oak and European beech and origin of their value.**

| Symbol | Description | Units | Value Sessile oak | European beech | Origin |
|---|---|---|---|---|---|
| **Carbon fixation** | | | | | |
| $k$ | extinction coefficient | m$^{-1}$ | 0.53 | | fitted with tree growth data of the study site |
| $PUE_{sl}$ | PAR use efficiency of sunlit leaves | kgC mol photons$^{-1}$ | 0.00006 | 0.000216 | fitted with tree growth data of the study site |
| $PUE_{sh}$ | PAR use efficiency of shaded leaves | kgC mol photons$^{-1}$ | 0.00105 | 0.000584 | fitted with tree growth data of the study site |
| **Respiration** | | | | | |
| $a_{sapwood}$ | parameters of the sapwood area function (Eq. 12) | | 0.00/1.54/0.16 | 0.00/0.00/0.52 | fitted with data from André *et al.* (2010) |
| $r_{npp\_gpp}$ | parameters of the npp to gpp ratio function (Eq. 8) | | 0.997/-0.386 | 0.959/-0.408 | fitted with tree growth data of the study site |
| $R_{Tref}$ | maintenance respiration per g of N at the reference temperature (15°C) | mole CO$_2$ gN$^{-1}$ h$^{-1}$ | 0.000079 | 0.000057 | fitted with tree growth data of the study site |
| $R_{gr}$ | growth respiration per unit biomass increment | kgC kgC$^{-1}$ | 0.2 | | Dufrêne *et al.* (2005) |
| $Q_{10\_leaf\ or\ fine\ root}$ | temperature dependence coefficient of leaf and fine root respiration | dimensionless | 2.1 | | Vose and Bolstad (1999) |
| $Q_{10\_stem\ and\ root}$ | temperature dependence coefficient of stem and root respiration | dimensionless | 1.7 | | Epron *et al.* (2001) |
| $Q_{10\_branch}$ | temperature dependence coefficient of branch respiration | dimensionless | 2.8 | | Damesin *et al.* (2002) |
| **Carbon allocation** | | | | | |
| $b_{leaf}$ | parameters of the leaf biomass function (Eq. 15) | kgC | 0.0026/1.96/1.96 | 19.04/1.30/0.00 | Jonard *et al.* (2006) |
| $b_{structural\_above}$ | parameters of the aboveground structural biomass (Eq. 26) | kgC | 0.000/263.4/0.969 | 0.056/292.8/0.966 | Hounzandj *et al.* (2015) and Genet et al. (2011) |
| $r_{root\_shoot}$ | root to shoot ratio | kgC kgC$^{-1}$ | 0.18 | | Genet *et al.* (2010) |
| $r_{fr\_leaf\_min}$ | minimum fine root to leaf ratio | kgC kgC$^{-1}$ | 0.5 | | literature data compilation |
| $r_{fr\_leaf\_max}$ | maximum fine root to leaf ratio | kgC kgC$^{-1}$ | 2.5 | | literature data compilation |
| $\delta_{leaf}$ | leaf relative loss rate | kgC kgC$^{-1}$ yr$^{-1}$ | 1 | | |
| $\delta_{fr}$ | fine root relative loss rate | kgC kgC$^{-1}$ yr$^{-1}$ | 1 | | Grote and Pretzsch (2002) |
| $f$ | stem form factor | m$^3$ m$^{-3}$ | 0.52 | | Hounzandj *et al.* (2015) and Genet et al. (2011) |
| $\rho$ | stem volumetric mass | kgC m$^{-3}$ | 562.17 | 556 | Hounzandj *et al.* (2015) and Genet et al. (2011) |
| $rt_{leaf}$ | leaf retranslocation rate | kgC kgC$^{-1}$ yr$^{-1}$ | 0.4 | 0.45 | determined based on tree foliage data collected in the study site |
| $rt_{root}$ | fine root retranslocation rate | kgC kgC$^{-1}$ yr$^{-1}$ | 0.4 | 0.45 | same values as leaves |
| $s_{branch}$ | parameters of the branch mortality function (same form as Eq. 15) | kgC | 5.50E-5/3.064/3.064 | 6.00E-4/2.681/0.00 | fitted with data from André et al. (2010) |
| $p_{fruit}$ | parameters of the fruit production function (Eq.22) | kgC | 1.55E-3/2.34 | 4.50E-4/2.681 | fitted with litterfall data from ICP Forests level II plots of Wallonia |
| $dbh_{threshold}$ | threshold dbh for fruit production | cm | 25 | | field observations |
| **Tree dimension increment** | | | | | |
| $hlce\%$ | fraction of the total height corresponding to the height of largest crown extension | m m$^{-1}$ | 0.81 | 0.77 | determined based on tree inventory data of the study site |
| $hcb\%$ | fraction of the total height corresponding to the crown base height | m m$^{-1}$ | 0.7 | 0.61 | determined based on tree inventory data of the study site |
| $Dd$ | parameters of the crown to stem diameter function (Eq. 10) | m m$^{-1}$ | 16.20/0.0280/0.00/0.00 | 10.49/0.00/1379/-2881 | determined based on tree inventory data of the study site |
| $sh$ | coefficient used to shift the mean crown to stem diameter ratio relationship to its maximum | dimensionless | 1.25 | 1.5 | determined based on tree inventory data of the study site |
| $r_{overlapping}$ | mean crown overlapping ratio | m m$^{-1}$ | 1 | 1.2 | determined based on tree inventory data of the study site |
| $\Delta hlce_{max}$ | maximum annual change in hlce | m yr$^{-1}$ | 0.5 | | determined based on tree growth data of the study site |
| $\Delta hcb_{max}$ | maximum annual change in hcb | m yr$^{-1}$ | 0.5 | | determined based on tree growth data of the study site |





**Table 3. Statistical evaluation of predicted basal area increments (vs observations) for various combinations of model options using normalized average error (NAE), paired *t*-test, regression test, root mean square error (RMSE), Pearson's correlation (Pearson's r) and modelling efficiency. Standard deviation or confidence intervals are provided in parentheses.**

| Model options | Mean basal area increment | | NAE | Paired t-test | orthogonal regression | | RMSE | Pearson's r | Modelling |
|---|---|---|---|---|---|---|---|---|---|
| Tree species | observations | predictions | | *P value* | intercept | slope | | | efficiency |
| *Castanea/npp to gpp ratio/distance-independent crown extension* | | | | | | | | | |
| European beech | 11.11 (13.27) | 10.92 (15.25) | -0.017 | 0.879 | 1.61 (0.58 - 2.63) | 0.87 (0.82 - 0.92) | 6.91 | 0.89 | 0.73 |
| Common oak | 16.94 (9.10) | 17.18 (11.92) | 0.014 | 0.865 | 3.80 (0.62 - 6.99) | 0.76 (0.61 - 0.92) | 8.58 | 0.69 | 0.10 |
| *Castanea/npp to gpp ratio/distance-dependent crown extension* | | | | | | | | | |
| European beech | 11.11 (13.27) | 9.36 (13.71) | -0.158 | 0.143 | 2.05 (0.92 - 3.19) | 0.97 (0.90 - 1.04) | 7.15 | 0.87 | 0.71 |
| Common oak | 16.94 (9.10) | 16.64 (11.70) | -0.017 | 0.839 | 4.00 (0.80 - 7.20) | 0.78 (0.62 - 0.93) | 8.51 | 0.69 | 0.12 |
| *Castanea/T°dependent maintenance respiration/distance-independent crown extension* | | | | | | | | | |
| European beech | 11.11 (13.27) | 11.57 (21.10) | 0.041 | 0.769 | 3.84 (2.42 - 5.26) | 0.63 (0.57 - 0.69) | 13.24 | 0.80 | 0.00 |
| Common oak | 16.94 (9.10) | 17.42 (14.59) | 0.029 | 0.768 | 6.07 (3.09- 9.05) | 0.62 (0.49 - 0.76) | 10.76 | 0.67 | -0.41 |
| *Castanea/T°dependent maintenance respiration/distance-dependent crown extension* | | | | | | | | | |
| European beech | 11.11 (13.27) | 13.16 (24.24) | 0.185 | 0.235 | 3.90 (2.62 - 5.18) | 0.55 (0.50 - 0.59) | 15.39 | 0.82 | -0.35 |
| Common oak | 16.94 (9.10) | 18.26 (16.24) | 0.079 | 0.459 | 6.67 (3.80 - 9.53) | 0.56 (0.44 - 0.68) | 12.11 | 0.68 | -0.79 |
| *PUE/npp to gpp ratio/distance-independent crown extension* | | | | | | | | | |
| European beech | 11.11 (13.27) | 8.66 (11.92) | -0.221 | 0.028 | 1.47 (0.37 - 2.57) | 1.11 (1.04 - 1.19) | 6.78 | 0.88 | 0.74 |
| Common oak | 16.94 (9.10) | 16.09 (9.71) | -0.05 | 0.511 | 1.83 (-2.80 - 6.46) | 0.94 (0.69 - 1.19) | 8.52 | 0.59 | 0.12 |
| *PUE/npp to gpp ratio/distance-dependent crown extension* | | | | | | | | | |
| European beech | 11.11 (13.27) | 6.61 (9.42) | -0.405 | <0.001 | 1.80 (0.51 - 3.08) | 1.41 (1.30 - 1.52) | 8.63 | 0.84 | 0.58 |
| Common oak | 16.94 (9.10) | 14.15 (9.08) | -0.164 | 0.026 | 2.73 (-2.63 - 8.09) | 1.00 (0.68 - 1.32) | 9.31 | 0.52 | -0.06 |
| *PUE/T°dependent maintenance respiration/distance-independent crown extension* | | | | | | | | | |
| European beech | 11.11 (13.27) | 7.57 (14.14) | -0.319 | 0.004 | 4.00 (2.35 - 5.65) | 0.94 (0.84 - 1.04) | 10.38 | 0.75 | 0.39 |
| Common oak | 16.94 (9.10) | 15.18 (10.28) | -0.104 | 0.185 | 3.49 (-0.40 - 7.37) | 0.89 (0.67 - 1.10) | 8.59 | 0.63 | 0.10 |
| *PUE/T°dependent maintenance respiration/distance-dependent crown extension* | | | | | | | | | |
| European beech | 11.11 (13.27) | 7.76 (14.49) | -0.301 | 0.007 | 3.99 (2.52 - 5.46) | 0.92 (0.83 - 1.00) | 9.75 | 0.79 | 0.46 |
| Common oak | 16.94 (9.10) | 14.59 (10.17) | -0.139 | 0.075 | 3.89 (-0.42 - 8.18) | 0.90 (0.65 - 1.14) | 9.15 | 0.58 | -0.02 |

