# Peer review of "HETEROFOR 1.0: a spatially explicit model for exploring the response of structurally complex forests to uncertain future conditions. I. Carbon fluxes and tree dimensional growth."

_Geoscientific Model Development, 2019_

## Referee Comment (RC1) · Anonymous Referee #1 · 30 Jun 2019

1. The authors present a new forest growth model HETEROFOR, which is a process-based model including comprehensive ecosystem and ecophysiological processes. As the title indicates, the inclusion of the competition of light for photosynthesis and structure growth between individual trees is one of the main contributions of this model. This manuscript focuses on the carbon budget and growth parts of the model. The processes behind them are well described in detail. The model was tested against inventory data and could predict the growth of tress very well. The authors further demonstrated the scenario simulations of future climate change using the model. In

general, this is a good manuscript and suitable for publication here.

2. In many places throughout the manuscript, the authors mentioned that there exists very few spatially explicit forest growth models (e.g. P4L22 "Given the lack of process-based models with detailed spatial representation..."). That's one of the motivations for developing this new HETEROFOR model. However, to my knowledge there are several excellent individual-based models owning this functionality, e.g. the iLAND model (http://iland.boku.ac.at/startpage) and the FORMIND model (http://formind.org/model/). The authors may want to survey the published models again and renew the manuscript.

3. In the last two paragraphs of the introduction, the scope as well as the strengths of the new HETEROFOR model are stated: it uses ray-tracing approach, hourly time step for calculating photosynthesis and transpiration, complex water balance module, detailed nutrient cycling module, and the CAPSIS platform. It would therefore be exciting for the readers, to explore these strengths by reading this paper. My major concern to this manuscript comes from this point: for what reason should the description of the whole model be separated into two (or more?) papers? I (and the readers) would like to know the main advances of this new model at once, instead of first knowing the carbon budget and growth parts, and waiting for the rest to come up later. The publication of a new and complex forest growth model has often been done in series papers, e.g. (Paper 1): full description of the model; (Paper 2) Verification, validation, and sensitivity analysis of the model; and (Paper 3) Application of the model. In the current manuscript of HETEROFOR, the general structure of the model is given in section 2.1 "Overall operation of the HETEROFOR model". However, the detailed description of water budget and nutrient cycling, and more importantly, their coupling to carbon budget are lacking. The RCP scenario analysis of the forest growth was performed and presented in the last part of this manuscript. This has been well done and the potential of the HETEROFOR model is well demonstrated. However, due to the lack of detailed model description on water and nutrient modules, we are actually not able to comment

on the outcome of this scenario simulations, or, even to give fair comments on the simulation results of individual radial growth and size-growth relationships.

4. If the manuscript should still be kept in its current scope, the authors may want to provide more information in section 2.1., including (a) the spatial resolution of the soil chemistry; (b) how the phenological periods are coupled to the ecosystem processes; (c) how does the stand-scale evaporation calculated by the Penman-Monteith method is distributed between soil, bark, and foliage. And how do the latter two compartments are further distributed to individual trees, such that the tree-scale throughfall and stemflow could be calculated? (d) how growth will be exactly regulated by the nutrient cycling? (e) Figure 1 should give an overview of the complete model, including the water and nutrient modules

5. P12L21: The HETEROFOR model takes into account 5 nutrients (N, P, K, Mg, Ca, in descending importance) in calculating the allocation of carbon in fine roots. How does the model deal with the weighting of the 5 nutrients?

6. In section 3.1, the npp of individual trees is compared with the modeled gpp. Please describe the method of calculating npp from the inventory data. On the other hand, why not directly compare the derived npp with the modeled npp?

7. When discussing the performance of calculating npp from gpp (P24), the authors focused only on the maintenance respiration. How about growth respiration?

8. P26L29: the term "threshold" is used here and in Figure 4 and is defined as "the minimum girth for radial growth to occur". I don't think that it is a proper way of description. The radial growth is too small to be properly displayed in the figure. However, the small trees do grow with girth smaller than the threshold

Some minor suggestions:

9. P3L6: propose -> proposed

10. P3L18-19: a stable systems -> a stable system

11. P3L21: short and long-term -> short- and long-term

12. P3L22: response -> responses

13. P3L33: horizontal dimension -> horizontal dimensions

14. P3L33: in both dimensions -> in three dimensions

15. P4L19: short -term -> short-term

16. P4L24: I am wondering if the authors are going to write: HETEROgeneous FORests?

17. P9L20: (The, 2006) -> (Teh, 2006)

18. P9L23: the extinction coefficient should be unitless

19. P17L27: LIEBL. -> Liebl.

20. P20L14: run with height different -> run with different

21. P23 Figure 5: use Sessile Oak instead of Common Oak for consistency

22. P29L4: NTOG 3D -> NOTG 3D

23. P44L28: Teh, C. -> Teh, C. B. S.

---

## Short Comment (SC1) · 4 Jul 2019

Thank you very much for taking the time to review our manuscript and for providing constructive comments and suggestions to improve it. We will consider them when revising the manuscript after receipt of the feedbacks from all reviewers.

Best regards, Frederic ANDRE

---

## Referee Comment (RC2) · Anonymous Referee #2 · 9 Jul 2019

The manuscript by Jonard et al. presents a new spatially explicit forest growth model which aims to incorporate the structural and/or compositional complexity in simulate forest growth, and associated ecological, biogeochemical, ecohydrological, physiological processes. This manuscript focuses on the carbon fluxes and tree growth and validates the model performance against inventory measurements at an oak/beech forest. The manuscript also demonstrates the potential applications by simulating forest growth based on several projected climate scenarios.

Overall, the manuscript is well written and carefully crafted. The study's objectives and

scopes are also generally justified. Below I list my few general comments. I would recommend the manuscript to be published in the Geoscientific Model Development after addressing/considering my general comments.

[1] This is a somehow very complicated model in terms of the number of parameters and model structures, but the model validation really falls short. The model predictions are tested against very sparse observation (i.e., two-time inventory at a single forest stand, only a few types of measurements). Given that this is a new and complicated model; I would question whether such validation is sufficient and robust. Strictly speaking, the comparison between gpp and npp isn't a valid comparison. Also, several parameters used (e.g., Table 2) are fit against the measurements at this specific site. I'd urge the authors should make a stronger case about the model robustness by considering a couple of options, e.g., test against more than one single sites, sites with different structure compositions, or multiple types of observations (e.g., those intermediate variables like respiration, leaf area, biomass). For observations that may be unavailable at the moment, the manuscript should at least point out the critical variables/parameters that need future data collection.

[2] I suggest considering removing or revising the parts of simulating forest growths based on projected climate scenarios. 1) The current model validation (as pointed out above) doesn't test the extrapolation capability of the model, e.g., either in time or under different climatic conditions. If the authors intend to keep the simulation part, they should consider/discuss those aspects in model validation. 2) For this manuscript, I think it may be a better idea to use simulations to demonstrate the capability or powerfulness of this model in simulating the spatially-explicit forest growths, e.g., simulations on forests with a different degree of heterogeneity in compositions/structures. I think it may help elaborate the reasons of why we need such type of model.

---

## Referee Comment (RC3) · Anonymous Referee #3 · 3 Aug 2019

This piece is sophisticated and well written. It may be publishable in GMD with some moderate revisions. However, the proposed model is not free and this gets on my nerves . . . a little bit. Open access has been the prevailing trend in academia and is good for science. There are many free codes available. If HETEROFOR is not free of charge, I am not sure the point of getting this piece published. I urge the authors considering to release the codes for the public.

My main comment is on the name of the model HETEROFOR, since I am not quite sure if the validation data is heterogeneous enough (only 2 broadleaf species). The

site was pretty homogenous to me comparing to the canopies in the tropical region. In addition, how representative is the validation site and data?

Here are my specific comments:

The abstract is a little bit disjointed. More information should be provided to clarify the sentences such as: Why the models called HETEROFOR and CAPSIS (acronyms for what), and how well is the radical growth prediction? Also, did you mention the effects of thinning in the abstract?

P2L19-20: These are not news and we know these all along. Why we need HETERO-FOR?

P3L14: "To explore forest response to new silvicultural practices . . ." Did you do that in this paper?

P5L12-13: "As the whole model could not . . ." Why is that? Please elaborate on it.

P10L30: I am not sure about eq. 6. Why the NPP/GPP ratio depends on the crown to stem diameter ratio?

P17L12-13: "Tree mortality occurs when trees reach a defoliation of 90%, considering that a tree with less than 10% of its leaves will never recover." Any reference for the statement?

P17L25-26: More detailed geographic and topographic information should be provided.

P18L13: Please justify the use of the Wallonia data.

P18L16: If the mean temperature of the site is 8 degrees C, why you used 15?

P20L6: Statistics to show no difference between the intercepts?

P23L10: CASTANEA

P46: Table 1 is not indexed in the ms.

---

## Author Comment (AC1) · 4 Sep 2019

**Final response to referee comments**

**Anonymous Referee #1**

1. The authors present a new forest growth model HETEROFOR, which is a process-based model including comprehensive ecosystem and eco-physiological processes. As the title indicates, the inclusion of the competition of light for photosynthesis and structure growth between individual trees is one of the main contributions of this model. This manuscript focuses on the carbon budget and growth parts of the model. The processes behind them are well described in detail. The model was tested against inventory data and could predict the growth of trees very well. The authors further demonstrated the scenario simulations of future climate change using the model. In general, this is a good manuscript and suitable for publication here.

Author response (R):

Thank you for these encouragements!

2. In many places throughout the manuscript, the authors mentioned that there exists very few spatially-explicit forest growth models (e.g. P4L22 "Given the lack of process-based models with detailed spatial representation: : :"). That's one of the motivations for developing this new HETEROFOR model. However, to my knowledge there are several excellent individual-based models owning this functionality, e.g. the iLAND model (http://iland.boku.ac.at/startpage) and the FORMIND model (http://formind.org/model/). The authors may want to survey the published models again and renew the manuscript.

R:

There are indeed several spatially explicit process-based models but some models account only for the radiation absorption (such as FORMIND) or describe the processes with an intermediate level of detail and simplified eco-physiological concepts such as the radiation use efficiency approach (e.g., iLAND). The FORMIND model was considered in the review of Pretzsch et al. (2015) but not retained according to our criteria (water balance and nutrient budget not accounted for). Regarding iLand, we added a sentence in the introduction to explain its specificity (description of the process at an intermediate level of detail in order to simulate forest dynamics at the landscape scale) and to explain the differences with our approach. We also rephrased some sentences such as that mentioned by the reviewer to be less categorical.

3. In the last two paragraphs of the introduction, the scope as well as the strengths of the new HETEROFOR model are stated: it uses ray-tracing approach, hourly time step for calculating photosynthesis and transpiration, complex water balance module, detailed nutrient cycling module, and the CAPSIS platform. It would therefore be exciting for the readers, to explore these strengths by reading this paper. My major concern to this manuscript comes from this point: for what reason should the description of the whole model be separated into two (or more?) papers? I (and the readers) would like to know the main advances of this new model at once, instead of first knowing the carbon budget and growth parts, and waiting for the rest to come up later. The publication of a new and complex forest growth model has often been done in series papers, e.g. (Paper 1): full description of the model; (Paper 2) Verification, validation, and sensitivity analysis of the model; and (Paper 3) Application of the model. In the current manuscript of HETEROFOR, the general structure of the model is given in section 2.1 "Overall operation of the HETEROFOR model". However, the detailed description of water budget and nutrient cycling, and more importantly, their coupling to

carbon budget are lacking. The RCP scenario analysis of the forest growth was performed and presented in the last part of this manuscript. This has been well done and the potential of the HETEROFOR model is well demonstrated. However, due to the lack of detailed model description on water and nutrient modules, we are actually not able to comment on the outcome of this scenario simulations, or, even to give fair comments on the simulation results of individual radial growth and size-growth relationships.

R:

We have long thought about the best way to present the HETEROFOR model. Given its complexity, it seemed to us that it was better to present separately the carbon-related processes from the water balance module. Therefore, I submitted the first paper on carbon flux and tree dimensional growth and my PhD student (Louis de Wergifosse) submitted the second paper. Both papers are under review in GMD and are already available as discussion paper which gives the opportunity to the reader to better understand the functioning of the water balance module. Given the length of the papers (49 equations in the first one and 83 in the second) and the fact that the second paper is part of the thesis of Louis de Wergifosse, it is not anymore possible for us to merge them in the same paper. Regarding the nutrient budget module, the implementation is not yet finalised and some tests must be performed. Given the complexity of the processes at play, it will require an article on its own. However, the model is fully functional without activating this module. It was therefore not considered in the scenario analysis.

4. If the manuscript should still be kept in its current scope, the authors may want to provide more information in section 2.1., including (a) the spatial resolution of the soil chemistry; (b) how the phenological periods are coupled to the ecosystem processes; (c) how does the stand-scale evaporation calculated by the Penman-Monteith method is distributed between soil, bark, and foliage. And how do the latter two compartments are further distributed to individual trees, such that the tree-scale throughfall and stemflow could be calculated? (d) how growth will be exactly regulated by the nutrient cycling? (e) Figure 1 should give an overview of the complete model, including the water and nutrient modules

R:

In the revised version of the manuscript, we provided the spatial resolution for the soil chemistry. We better explained how phenology interacts with the other processes and how the evaporation is distributed between soil, bark, and foliage based on the absorbed solar radiation by each ecosystem compartment. As bark and foliage are calculated at the stand scale as well as throughfall and stemflow, they are not distributed to individual trees. We did not describe with more details how tree growth is regulated by nutrient availability since the complexity of the nutrient cycling and tree nutrition module really disserves a paper on its own. Accordingly, we have integrated the phenology and water balance in Figure 1 but not the nutrient cycling in order to keep it readable. In the paper of de Wergifosse et al. currently in discussion in GMD, a figure describing the water balance module with more details is provided.

5. P12L21: The HETEROFOR model takes into account 5 nutrients (N, P, K, Mg, Ca, in descending importance) in calculating the allocation of carbon in fine roots. How does the model deal with the weighting of the 5 nutrients?

R:

For each nutrient, a fine root to foliage ratio is calculated and the maximum is retained in order to account for the fact that the most limiting nutrient has the dominant effect. In the revised manuscript, we have completed our description to make this point clearer.

6. In section 3.1, the npp of individual trees is compared with the modeled gpp. Please describe the method of calculating npp from the inventory data. On the other hand, why not directly compare the derived npp with the modeled npp?

R:

The method used to calculate npp from the inventory data is described in section 2.2.7 on growth reconstruction. In the revised version of the manuscript, we provided more details on this method in order to make it perfectly clear for the reader.

To transform the modelled gpp into npp, we used either a ratio or the routine for respiration calculation. In both cases, we used a parameter fitted based on tree growth data. In order to keep the two variables completely independent, we did not transform the modelled gpp into npp.

7. When discussing the performance of calculating npp from gpp (P24), the authors focused only on the maintenance respiration. How about growth respiration?

R:

In HETEROFOR, the growth respiration is estimated as a proportion of the total biomass increment as in many other models (e.g., 3D-CMCC, CASTANEA). The growth respiration is therefore simpler to estimate than the maintenance respiration and the challenge for estimating growth respiration is to correctly estimate the biomass increment.

8. P26L29: the term "threshold" is used here and in Figure 4 and is defined as "the minimum girth for radial growth to occur". I don't think that it is a proper way of description. The radial growth is too small to be properly displayed in the figure. However, the small trees do grow with girth smaller than the threshold.

R:

We agree with the reviewer that trees smaller than the threshold still grow even if this growth is very limited. We redefined the threshold as the girth beyond which radial growth linearly increases with girth.

Some minor suggestions:

9. P3L6: propose -> proposed

10. P3L18-19: a stable systems -> a stable system

11. P3L21: short and long-term -> short- and long-term

12. P3L22: response -> responses

13. P3L33: horizontal dimension -> horizontal dimensions

14. P3L33: in both dimensions -> in three dimensions

15. P4L19: short -term -> short-term

16. P4L24: I am wondering if the authors are going to write: HETEROgeneous FORests?

17. P9L20: (The, 2006) -> (Teh, 2006)

18. P9L23: the extinction coefficient should be unitless

19. P17L27: LIEBL. -> Liebl.

20. P20L14: run with height different -> run with different

21. P23 Figure 5: use Sessile Oak instead of Common Oak for consistency

22. P29L4: NTOG 3D -> NOTG 3D

23. P44L28: Teh, C. -> Teh, C. B. S.

R:

All minor suggestions were accepted and implemented accordingly in the revised manuscript.

**Anonymous Referee #2**

The manuscript by Jonard et al. presents a new spatially explicit forest growth model which aims to incorporate the structural and/or compositional complexity in simulate forest growth, and associated ecological, biogeochemical, ecohydrological, physiological processes. This manuscript focuses on the carbon fluxes and tree growth and validates the model performance against inventory measurements at an oak/beech forest.

The manuscript also demonstrates the potential applications by simulating forest growth based on several projected climate scenarios.

Overall, the manuscript is well written and carefully crafted. The study's objectives and scopes are also generally justified. Below I list my few general comments. I would recommend the manuscript to be published in the Geoscientific Model Development after addressing/considering my general comments.

[1] This is a somehow very complicated model in terms of the number of parameters and model structures, but the model validation really falls short. The model predictions are tested against very sparse observation (i.e., two-time inventory at a single forest stand, only a few types of measurements). Given that this is a new and complicated model; I would question whether such validation is sufficient and robust. Strictly speaking, the comparison between gpp and npp isn't a valid comparison. Also, several parameters used (e.g., Table 2) are fit against the measurements at this specific site. I'd urge the authors should make a stronger case about the model robustness by considering a couple of options, e.g., test against more than one single sites, sites with different structure compositions, or multiple types of observations (e.g., those intermediate variables like respiration, leaf area, biomass). For observations that may be unavailable at the moment, the manuscript should at least point out the critical variables/ parameters that need future data collection.

R:

Individual gpp is obtained with the photosynthesis routine of CASTANEA based on the intercepted radiation while the reconstructed npp is obtained from the observed increment in dbh and height. The tree growth measurements are just used to calculate the reconstructed npp. For the gpp, the CASTANEA model was calibrated independently on other sites. In conclusion, we consider that the two variables are independent.

The first objective of the paper was to present a description of the model and secondary to make a first evaluation of the model using an uneven-aged and mixed stand. We are well aware that this first evaluation is insufficient and we are preparing a larger one at the European level using ICP forest level II plots (cf. Conclusion and future prospects). For the present article, we will expand this first evaluation to 3 stands covering contrasted stand structures and compositions. For this first evaluation, we will use only the tree growth measurements. However, the companion paper which focus on phenology and water cycle (de Wergifosse et al., in review in GMD) is using phenological data as well as throughfall, stemflow and soil water content measurements for the model evaluation. For our evaluation at the European scale, we could eventually also use litter fall data and eddy covariance data on some sites of the ICOS program.

[2] I suggest considering removing or revising the parts of simulating forest growths based on projected climate scenarios. 1) The current model validation (as pointed out above) doesn't test the

extrapolation capability of the model, e.g., either in time or under different climatic conditions. If the authors intend to keep the simulation part, they should consider/discuss those aspects in model validation. 2) For this manuscript, I think it may be a better idea to use simulations to demonstrate the capability or powerfulness of this model in simulating the spatially-explicit forest growths, e.g., simulations on forests with a different degree of heterogeneity in compositions/structures. I think it may help elaborate the reasons of why we need such type of model.

R:

In the revised version of the manuscript, we have evaluated HETEROFOR in 3 stands showing contrasted species composition and stand structures, which is in line with the second suggestion of the reviewer. For the part with the simulations based on projected climate scenarios, we agree to remove it if necessary. However, we suggest to keep it as we think this is a good illustration of the potential of HETEROFOR (see comment n°3 of reviewer 1). Now that the companion paper is also accessible in GMD, the reader has all the information to interpret it. Indeed, this second paper describes processes that are sensitive to climate conditions (phenology, water balance) and evaluates them against observations.

**Anonymous Referee #3**

This piece is sophisticated and well written. It may be publishable in GMD with some moderate revisions. However, the proposed model is not free and this gets on my nerves : : : a little bit. Open access has been the prevailing trend in academia and is good for science. There are many free codes available. If HETEROFOR is not free of charge, I am not sure the point of getting this piece published. I urge the authors considering to release the codes for the public.

R:

HETEROFOR is actually free and we decided to distribute it freely. The misunderstanding probably comes from our wording which was not clear enough and from the fact that we did not choose any specific license. In CAPSIS, the models have by default no license and the rights belong to the model authors. In order to further clarify it, we decided to adopt a LGPL license for HETEROFOR.

We have now clarified the wording regarding the code availability section and mentioned that a LGPL license was adopted for HETEROFOR.

My main comment is on the name of the model HETEROFOR, since I am not quite sure if the validation data is heterogeneous enough (only 2 broadleaf species). The site was pretty homogenous to me comparing to the canopies in the tropical region. In addition, how representative is the validation site and data?

R:

Compared to forest in tropical region, the mixed broadleaved stand is indeed less heterogeneous. For temperate forests, it is however already quite heterogeneous (especially when compared to monospecific even-aged stands). In this stand, part of the heterogeneity comes from the size class distribution (see below).

The idea was just to make a first evaluation of the model performances using a mixed and uneven-aged stand of oak and beech. A broader evaluation using stands from the whole Europe is currently done and the results will be available later.

Nevertheless, we decided to expand this first evaluation and use 3 stands with contrasted stand structure and composition.

[Figure]

Here are my specific comments:

The abstract is a little bit disjointed. More information should be provided to clarify the sentences such as: Why the models called HETEROFOR and CAPSIS (acronyms for what), and how well is the radical growth prediction? Also, did you mention the effects of thinning in the abstract?

R:

In the revised version of the abstract, we defined the acronyms HETEROFOR (*HETEROgeneous FORest*) and CAPSIS (*Computer-Aided Projection of Strategies In Silviculture)* and provided the Pearson's correlation between observed and predicted radial growth. We did not mention the effects of thinning since we did not test it in this paper.

P2L19-20: These are not news and we know these all along. Why we need HETEROFOR?

R:

We removed this sentence from the manuscript and explained later in the introduction the specificity of HETEROFOR.

P3L14: "To explore forest response to new silvicultural practices : : :" Did you do that in this paper?

R:

No, because the manuscript is focused on the model description but we developed HETEROFOR with this goal and are currently using HETEROFOR in this way in several projects.

P5L12-13: "As the whole model could not : : :" Why is that? Please elaborate on it.

R:

This sentence was removed from the introduction and the explanation regarding the model description is given at the end of section 2.1.

P10L30: I am not sure about eq. 6. Why the NPP/GPP ratio depends on the crown to stem diameter ratio?

R:

We realized that, for some tree species, the NPP/GPP ratio varies with tree characteristics and observed a clear effect of the crown to stem diameter ratio regarding oak whose crown development strongly depends on the stand density (and past silvicultural treatment). This possibility was therefore integrated in the model. However, if the user does not want to use it, the corresponding parameter ($\beta$) can be fixed to 0 (Eq. 8).

P17L12-13: "Tree mortality occurs when trees reach a defoliation of 90%, considering that a tree with less than 10% of its leaves will never recover." Any reference for the statement?

R:

In the revised version of the paper, we rephrased it "…considering that a tree with less than 10% of its leaves is in an advanced stage of decline and is unlikely to recover (Manion, 1981)." and we added a reference.

Manion, P.D., 1981. Tree Disease Concepts. 1st Edn. Prentice-Hall, Englewood Cliffs, NJ, 402 pp.

P17L25-26: More detailed geographic and topographic information should be provided.

R:

We added in the revised version that the stands are located on a tableland. We also provided the geographic coordinates, the altitude and the region in which the stands are located.

P18L13: Please justify the use of the Wallonia data.

R:

We used data collected in the ICP Forests level II plots of Wallonia only for fitting the parameters of the fruit production equation. We chose the plots from Wallonia to have the same ecological conditions than the study site and since we had access to this data. However, the user can provide a file with fruit production data that will be used by the model to adapt the parameters of the fruit production equation.

P18L16: If the mean temperature of the site is 8 degrees C, why you used 15?

R:

15°C is the reference temperature at which the parameter is evaluated (the maintenance respiration per g of N at 15°C) but, in Eq. 11, the temperature effect is taken into account which enables to calculate the maintenance respiration at all temperatures.

P20L6: Statistics to show no difference between the intercepts?

R:

In Figure 2, the values provided between parentheses are half the confidence intervals. Since the confidence intervals of the intercepts overlap, we considered that the difference was not significant.

P23L10: CASTANEA

R: OK

P46: Table 1 is not indexed in the ms.

R: OK

---

## Author Response (AR2)

**Response to referee comments**

The revised manuscript with track changes is provided below.

**Anonymous Referee #4**

Suggestions for revision or reasons for rejection (will be published if the paper is accepted for final publication)

I very much appreciate the development and implantation of a new individual tree model that is physiology based and can be applied over periods that allow to judge the development of species mixtures under changing environmental conditions. I feel however that considerable improvements in the model description, presentation and discussion could still be achieved.

 Model description

 The model description in general would also benefit from a better explanation about which processes are run in which time steps. While photosynthesis is calculated hourly, time steps for allocation are not explicitly addressed. They may be done annually, similarly to dimensional calculations. If this is true (I couldn't find any statement about it), it raises some yet unexplained questions: Where is the foliage compartment outside the vegetation period and how does it respire? Is the biomass at the end of the year abruptly changing or are empirical functions used to distribute the growth over the year? Shouldn't the fact that biomass is actually changing during the year be reflected in the maintenance respiration calculations? If, however, allocation is done hourly or daily, how is the first leaf flushing justified in deciduous trees when there is not yet any photosynthesis? In particular since there seems to be no reserve compartment. The best model description is in the caption of figure 1 but I think this (and more) information needs to be in the text too.

Author response (R):

We reviewed the section 2.1 of the paper "Overall operation of the HETEROFOR model" and we specified the time step of each process. Carbon allocation is done once a year at the end of the vegetation period which allows to update tree dimensions for the next yearly time step during which tree size does not change. Outside the vegetation period, the foliage compartment is not considered and does not respire. The small respiration rate of buds during the dormant period is therefore not explicitly accounted but is indirectly taken into account by the calibration of the respiration coefficient.

 I also noticed that in many equations, parameters are given as alpha, beta, gamma and delta letters without further indication - which is a bit strange and sincerely against common rules since this kind of notation is not specific. In my opinion, also empirical parameters (including a, b, c...) should be unmistakably identifiable and should also be given and explained (including sources) in a table somewhere.

R:

The value of these parameters is given in Table 2. We recognize however that this information did not appear sufficiently clearly. In the revised Table 2, we clearly identify the parameters, their value and the associated equation (changes highlighted in red).

 Model demonstration and discussion

I appreciate that examples for demonstrating the model's performance are based on a number of stands that differ somewhat in their structure. I was a bit disappointed though that only different model approaches (such as different photosynthesis schemes) have been compared and no parameter sensitivity has been done. I admit, however, that this might be a very comprehensive demand and should possibly be restricted to completely new model formulations. On the other hand, the tests are still limited to two tree species and more or less only one climate area. If no other sites will be incorporated (which would be good but which is something I don't really expect), the issue should be carefully acknowledged as a limited demonstration of model performance.

R:

The objective of this paper is first to describe a new model and then to show that it gives interesting results in some case studies. We agree that this first model evaluation is limited and have better acknowledged this in the conclusion (P36L10-11). This first evaluation will be complemented by a larger one at the European scale.

In addition, existing literature should be better considered for the following topics: 1) the linear relationship between NPP and GPP has been recently discussed by Collalti et al. 2019; 2) the allometry depends on competition issues that need to be treated based on individual tree situations, which is a likely explanation for the relatively bad performance of increment calculations (see e.g. del Rio et al. 2019), 3) an hierarchical allocation approach that distributes carbon into different compartments has certain disadvantages, so that source-sink approaches or other approaches needs to be at least discussed as possible alternatives (e.g. Carl et al. 2018, Thurm et al. 2017). These points are particularly important in in a single-tree model than in a stand model.

R:

The references suggested by the reviewer helped us to substantially improve the discussion. Collalti et al. (2019) was used to discuss the pros and cons of the *npp* to *gpp* ratio approach for modelling autotrophic respiration (P31L8-25). The way our model accounts for the impact of intra and inter-specific competition on tree allometry was analysed in the light of the study of Del Rio et al. (2019) (P31L34 to P32L3, P33L4-6). The study of Carl et al. (2018) allowed us to improve the discussion on the size-growth relationships and on the competition mode (P33L18-33) while improvement perspectives were opened thanks to Thurn et al. (2017) (P33L15-16).

Besides the evaluations which are basically based on growth measurements that are very integrated outcomes from many underlying processes, I would like to see some curves of how different compartments of different trees or groups of trees are developing over time in dependence on climatic input. Do we see any climate or management impact here? Are different groups have different strategies or are developing differently? Is this development reasonably at all? Please elaborate.

R:

We followed the reviewer suggestion and carry out one-year simulations to highlight the impact of climate conditions (comparison of a dry, normal and wet year) and of management (reduction of stand basal area by 25%) on biomass production and on its allocation to tree compartments. This new simulation experiment was described in the material and methods (P21L17-24), presented in the results (P26L6-16 + Fig. 5) and discussed (P33L7-16).

There are a couple of other more specific things that caught my eye and may be corrected or answered:

P3L13: Another argument why forestry trials are not suitable to derive estimates for future conditions are that these future conditions, i.e. CO2 concentrations, simply cannot be observed in the present.

R:

Thank you for the suggestion! We have considered it in the revised version (P3L12).

P4L13ff: The BALANCE model is described here as having no soil layers and is missing basic soil processes. This is wrong, as can be easily derived from Figs. 3 and 6 in Grote and Pretzsch 2002. I admit that neither weathering nor nutrients other than nitrogen are included but still the model hosts a full carbon and nitrogen cycle and calculations in different soil layers, considering different nutrient availability with rooting depth. Please correct.

R:

In the revised version, we corrected the description of BALANCE which considers indeed different soil layers (P4L13-14).

P6L14: Does the model run when 'no meteorological measurements' (resp. scenario data) are provided as input? How can this work out reasonably?

R:

When no hourly meteorological measurements are available, the model can be run with a default empirical approach based on the radiation use efficiency approach. In this case, the only input needed by the model is the mean monthly global radiation. This more empirical approach provides good results for conditions similar to those in which the model was calibrated but do not allow extrapolations to future conditions. This option does not account for inter-annual variations in tree growth.

P7L12ff (also in Fig.1, and P9): GPP is either derived by a radiation efficiency approach or with the Farquhar model? This is irritating. There should be a rule when the one or other method is applied. Has this something to do with the age/size or position of the tree similar to the differentiation of respiration?

R:

The user can choose one of the two options (radiation efficiency approach or Farquhar model) depending on his objectives and on meteorological data availability. We rephrased it to avoid misunderstanding (P7L23-25, legend of Fig. 1).

P7L19: The expression 'compartment' or 'tree compartment' is used here and in other places but never defined. In P6L32 it is used for bark and foliage, P8L1 it is soil solution, at P12 it describes foliage and fine roots, and at P26 we learn that 'structural compartments' exist besides foliage and roots. The same holds for the expression 'organ' which is partly used as a synonym to 'compartment'. Figure 1 caption tells us about 'structural organs' that are 'roots, trunk, and branches' and also later 'organ' development is described with equations throughout P11 to P13 without telling the reader what is actually meant.

R:

In order to clarify the scientific terms, we defined 'tree compartment' in P6L11-12 and did not use anymore the term 'organ'. We also restricted the use of the word 'compartment' to 'tree compartment' and modified the sentence where it was used for the soil solution.

 P9L7 (and figure 1): Fruits are very interesting but seldomly considered as a separate and dynamic tree compartment that is explicitly modelled. From Fig. 1 and the description in P13 it seems that it is considered but how data on this compartment can actually be used to modify allocation pattern, needs to be described further (taking the carbon from all other compartments equally or selectively?).

R:

As for the other functional tree compartments (foliage, fine roots), the carbon necessary for fruit production is subtracted from npp before the allocation to the structural tree compartments (Eq. 23).

 P9L10: This indicates that photosynthesis is driven with daily meteorological data which for the Farquhar approach only makes sense if data are downscaled to hourly (or similar) time steps (also indicated in P10L18). Why is this not possible for the hydrological model then which explicitly requires hourly input?

R:

Hourly (and not daily) meteorological data are used to calculate photosynthesis with the Farquhar approach as mentioned in P11L18.

 P10L14: Possible relationships between Farquhar photosynthesis and soil water restrictions have been implemented before (see e.g. Granier, Knauer, van Wijk, Wang in references). I guess, that also the new implementation will be based on one of them which could be indicated here. Otherwise, I fear that a short description of how this is done is indispensable for understanding the carbon assimilation dynamics.

R:

In HETEROFOR, the impact of soil water on Farquhar photosynthesis is described with a decreasing exponential function of the soil water potential. This relationship is presented in the companion paper (see Eq. 56 in de Wergifosse et al., in review).

P11L4ff: I am a bit puzzled about these explanations. Why is it bad that 'the crown to stem diameter ratio changes during the course of the tree development' so that the npp/gpp ratio also changes. Isn't this exactly what you want? And why is this index better although a relation of npp/gpp to this index has not been demonstrated?

R:

The crown to stem diameter ratio characterizes the tree shape reflecting past competition conditions but also changes during the course of the tree development. We standardized it to remove the size effect in order to obtain an index only dependent on the tree shape. This index is particularly useful to account for the large difference in oak crown extension according to the silvicultural system (large crowns in former coppices with standards *vs* narrow crowns in dense high forests). For oak, we observed that this index provides good results. However, the user can fix the parameter associated to this index to 0 for some tree species (we did for beech) or if he is not convinced by its added value. Fixing this parameter to 0 is equivalent to use a constant npp/gpp ratio.

We rephrase this paragraph in order to clarify the explanations (P12L1-11).

P11L16ff: Maintenance respiration is taken from photosynthesis (according to equation 7), but its calculation is independent of carbon gain. So, the net gain might get negative. What's happening then? Is there any reserve compartment?

R:

The net primary production (npp) is calculated once a year by subtracting maintenance respiration from gross primary production (Eq. 7) and no reserve compartment is considered. Contrary to what frequently happen when npp is calculated daily, the net gain is nearly always positive when calculated on an annual basis. If it turns out that the npp is negative due to exceptional conditions (highly outcompeted trees, water stress), the tree dies. Generally, when it happens, npp decreases progressively and tree decline takes several years.

P13L15/Eq.22: How is this parameterized and even how is the function itself derived? My question arises from the fact that fructification is highly variable and relates to specific climate conditions such as temperature in spring. Also, the parameter given in the appendix seems to be a variable based on this equation.

R:

The inter-annual variation in fruit production is not described by a specific routine. For past conditions, the user can provide an optional input file with the fruit production per year and tree species. For future or unknown conditions, mean value are used by default. The fruit distribution among trees depends on their size and on the light availability according to Eq. 22. In this equation, the parameter α takes a default value or is adapted based on the fruit production of the year (provided in the input file). In the future, we plan to develop a more sophisticated routine to predict the inter-annual variations in fruit production and to distribute it among trees.

In the revised version of the paper, we provided additional explanations on the fruit production modelling (P14L20-24)

P13L30ff/Eq.26-28: How could height and diameter growth be derived from the total above structural carbon gain if diameter and height growth only depend on stem (trunk) development? I would expect that the dimensional growth depends on db_stem rather than db_structural. Given

that the derivation of height and diameter increment is correct, has anybody checked if the biomass increase is actually supporting the dimensional change? Or does it imply a wood density change?

R:

Height and dbh growth are obtained from the aboveground structural biomass gain using an allometric equation predicting aboveground structural biomass from height and dbh (Eq. 26). To do so, this allometric equation was derived with respect to time and then rearranged (Eqs 27 and 28).

The increment in stem biomass is not used to calculate dbh and height since it is still unknown at this stage of the calculation. It is calculated once the dbh increment is known (Eq. 34). As the dimensional change is calculated based on the aboveground structural biomass gain, this biomass increase perfectly corresponds to the dimensional change, without implying any change in wood density. Carbon budgets were achieved to check that no calculation errors were made.

P14L30: Hounzandji et al. 2015 distinguish branches/twigs but not roots into three fractions of different diameter classes. The assumption that roots could be described with the same equations (and parameters?) without any additional data seems very bold to me. As it is, the description gives a false impression of what has been derived from literature. Furthermore, if a root fraction of 0-4 cm is assumed, it would include fine roots, which are, however, separately treated in the model. The differentiation is therefore inconsistent.

R:

The equations of Hounzandji et al. (2015) provide branch proportions and not absolute values. We considered that the mathematical forms of the equations is also suitable for roots as long as the parameters are adjusted with root data. Since we did not have such data, we temporarily used the same parameters than for the branches. However, these distributions in root categories has no impact on the functioning of the model since this information is not used elsewhere. These are just model outputs. If the user does not trust this categorization, he may just consider the whole root compartment without distinguishing it by category. As the proportions are applied to the root compartment, the fraction < 4 cm does not include the fine roots. This is the reason why it is called '< 4 cm' and not '0 - 4 cm'. In the revised version of the paper, we clarified it (P16L6-9).

P17L17ff: The reconstruction is a nice feature but depends on the knowledge about how many trees have died and when they died during the investigated period. Otherwise, npp is underestimated since it only refers to the remaining trees. Could you acknowledge this in the paragraph?

R:

In addition to repeated stand inventories, the reconstruction tool also requires a file listing the trees which were cut or died between the two inventory dates and the last year during which they were present in the stand. We added this information in the revised version (P19L26-28).

[revised manuscript text omitted]

---

## Author Response (AR3)

**Response to referee comments**

The revised manuscript with track changes is provided below.

**Suggestions for revision or reasons for rejection (will be published if the paper is accepted for final publication)**

Thank you for considering some remarks and answering a number of questions raised before. I particularly appreciate the additional simulations.

Although I am still a bit concerned that the various options of the model (with and without water balance, LUE or Farquhar model, various alternatives in determining height growth and crown dimensions) may require more detailed instructions and evaluations, I acknowledge that this is difficult to carry out in a paper. Perhaps the authors can provide a commenting sentence?

Response (R): Among the various options listed by the reviewer, we compared the two alternative options for calculating GPP (PUE *vs* Farquhar model) and also for describing crown extension (distance-dependent *vs* -independent approach). The water balance was activated for all the simulations. Some option combinations were therefore not tested, such as the PUE approach without activating the water balance. We added two sentences to explain this (P20L14-21).

Another issue that concerns me is that the nutrient cycle, a feature that makes the new model to some degree special, seems to affect respiration and allocation but not photosynthesis. Does this mean that with less nutrients the overall biomass growth is increasing? This would not only be counter-intuitive but also against experiences (e.g. Weinstein et al.).

R: In HETEROFOR, tree mineral nutrition has a direct impact on photosynthesis. Indeed, the nitrogen foliar concentration is used to estimate the maximal carboxylation rate in the photosynthesis routine of CASTANEA (Dufrêne et al., 2005). In the revised version of the manuscript, we clarified this in *2.2.1 Initialization* (P10L10-13) and in *2.2.2 Gross primary production* (P11L30-31).

In addition, while going through the text again I stumbled about a number of smaller issues and phrases that are uncommonly worded and difficult to understand, particularly in the new parts. This is certainly not covering any language problem and a general check is encouraged.

P1L15: I think you mean 'different' instead of 'contrasted'

R: OK, we made the change (P2L15).

P1L18: The "most empirical option for describing maintenance respiration provides the best results while … empirical approaches similarly perform for photosynthesis and crown extension." What should the reader who cannot distinguish between 'empirical' and 'most empirical' gain from this? Rephrase.

R: OK, we rephrased this sentence (P2L18-20).

P1L20: 'driven by' instead of 'according to'? (also to avoid repetition)

R: OK, we made the change (P2L21-22).

P2L4: rephrase 'make no doubt' do you mean 'will certainly happen' or similar?

R: OK, we rephrased this sentence to clarify its meaning (P3L4-5).

P2L6: Please indicate what Messier had proposed.

R: Messier proposed another vision of the forests considered as complex adaptive systems. We clarified the sentence (P3L7).

P2L12: After modification, this sentence doesn't make any sense now. Rephrase.

R: OK, we rephrased it (P3L13-14).

P2L14ff: Pretzsch et al. don't propose new models but a test of the (relatively few) models that are able to cover complex stands with empirical data. Reformulate!

R: we replaced the reference of Pretzsch et al. (2015) by two others more appropriate (Berger et al., 2007; Bravo et al., 2019) => P3L17-18.

P4L33: 'within the frame of' instead of 'in'

R: OK (P5L1).

P5L1ff: Repetition from P2L8ff

R: We deleted some part of the abstract to avoid redundancy (P2L9-13).

P6L4: grammar, rephrase

R: OK (P6L4-5).

P6L6: Is it really only the annual time step that is stored and can be used for evaluation? Shouldn't there be the option to read out at least daily data, e.g. for checking photosynthesis and drought development? Since later on 'detailed budgets' are mentioned, I think that the information should be elaborated.

R: Some variables (foliage state, water fluxes, npp and gpp) are stored at an hourly or daily time step in java objects created annually. This information is accessible to the user through exports (see user manual). This information is now provided in the manuscript (P6L7-9).

P6L24ff: Repetition from the previous sentence. Shorten and rephrase.

R: OK (P6L25-30).

P7L34: 'On another hand'? Wrong phrase (perhaps consider 'In addition' or just skip?)

R: OK, we rephrased this sentence (P8L3-4).

P10L11: I am still puzzled. Why is the hourly meteorology only loaded for water-balance calculations? It is needed for photosynthesis anyway, correct? If this is linked to the selection of the photosynthesis module, please note this here.

R: We clarified this point (P10L16-19).

P11L15: How is the position of the leaf accounted for? Simple linear decreasing function? Exponential function using two additional parameters according to Schaefer et al.? Which parameters? Same for all species?

R:  In Eq. 56 in de Wergifosse et al. (in review), stomatal conductance is inversely proportional to the height of maximum crown extension. This information was added P11L21-22.

P14L4: 'Allocation priority is given….' Not 'In the allocation, priority is given …'

R: OK (P14L9).

P14L19: Either you skip the new half-sentence or you have to give a reference here. Perhaps look into Hacket-Pain et al.

R: We provided two references to support this statement (Greene et al., 2002; Davi et al., 2016) => P14L30.

P16L12: described not describe

R: OK (P16L21).

P19L31: grammar, rephrase

R: OK (P20L8-9).

P20L17ff: I may have missed it but after highlighting the special consideration of nutrients for growth in HETEROFOR, it should somewhere be stated explicitly that the state of nutrition has been kept constant for all presented simulations.

R: OK, we mentioned it at P20L11-12.

P20L19: replace 'evolution' by 'development'

R: OK (P20L31).

P21L8/9: grammar, rephrase

R: OK (P21L21-23).

P23L3: what does 'but not symmetrically' mean? (also correct spelling)

R: We mean "in all cases" (systematically). => (P23L3)

P24L4 (Fig.4): I wonder why no simulation vs. measurement plots but only simulation and measurement vs. girth?

R: We presented simulation vs measurement plots in Fig. 3.

P25L6,L12,L13,L15: grammar, rephrase

R: OK (P25L6-20).

P25L17ff: This information is very scarce and provokes some serious question. In order to judge the reason for the insensitivity to the climate scenarios, the reader needs to know in particular, if the water availability was different. The role of water balance is mentioned in the discussion referring to another paper. Still, I think that some information of increasing drought stress or vegetation length extension could be indicated here in the result section.

R:
The apparent insensitivity of tree radial growth to climate scenarios (when $CO_2$ concentration is kept constant) is explained by two opposite trends (the increased in vegetation period which has a positive impact and the increased in drought stress which has a negative effect). These aspects are mentioned in the discussion and analysed in detailed in de Wergifosse et al. (in review b).
As the present paper is already quite long (35 pages, 6 figures and 3 tables), we cannot increase it yet with results and tables/ figures on drought stress and vegetation period length, especially since this information is in another paper. This paper was however not referenced correctly in the previous version. We have now corrected it.

P25L18: 'recognized' instead 'assessed'?

R: OK (P25L20).

P30L1: 'increase' instead 'increased'

R: OK (P30L1).

P30L10: 'maintain a stable npp to gpp ratio', 'On the other hand' (never: on another hand – however, this phrase is usually only used if another sentence before has been started with: On the one hand, …)

R: OK (P30L10).

P30L28: 'evaluate the interest of this approach'?? Do you mean the significance, applicability, …?

R: we mean "relevance" and corrected accordingly (P30L28).

P31L8: This is an unproven claim. You might however phrase it as a possible line of further investigations.

R: We deleted this sentence (P31L7-8).

P31L16: provide instead provides

R: OK (P31L16).

P31L20: 'high computational demand' instead 'long computing times'

R: OK (P31L20-21).

P31L24: rephrase 'two errors were added'

R: We rephrased this sentence (P31L24-26)

P31L28: delete 'noisily'

R: OK (P31L29).

P31L29: 'depict' instead of 'highlight'?

R: OK (P31L30).

P32L3: 'account' instead of 'accounts'

R: OK (P32L3).

P32L5: replace 'What came out' by 'The result'

R: OK (P32L6).

P32L9,L12: grammar, rephrase

R: OK (P32L10-11).

P32L13: Could? Or Should? Or Can?

R: We mean "could" and rephrased the sentence (P32L13-14).

P33L27: I don't see the logical reasoning here. Also avoid to describe coupling as 'perfect'.

R: We deleted "perfect" (P33L29-30).

P34L4/5: grammar, rephrase

R: OK (P34L4-6).

P34L7: I would delete 'predicts well individual radial growth and' for language reasons as well as because I think that this feature has not been convincingly demonstrated (see comment to Fig. 4, also, prediction needs longer periods and independently set up sites).

R: OK (P34L9).

P24L10: replace 'overview' by 'impression'

R: OK (P34L12).

Mentioned references
Hacket-Pain, A., Ascoli, D., Berretti, R., Mencuccini, M., Motta, R., Nola, P. et al. 2019 Temperature and masting control Norway spruce growth, but with high individual tree variability. Forest Ecol. Manage., 438, 142-150.
Weinstein, D.A., Beloin, R.M. and Yanai, R.D. 1991 Modeling changes in red spruce carbon balance and allocation in response to interacting ozone and nutrient stresses. Tree Physiol., 9, 127-146.

[revised manuscript text omitted]